# Bacterial motility can govern the dynamics of antibiotic resistance evolution

Vit Piskovsky [1,2] & Nuno M. Oliveira [1,3] ✉

Spatial heterogeneity in antibiotic concentrations is thought to accelerate the evolution of antibiotic resistance, but current theory and experiments have overlooked the effect of cell motility on bacterial adaptation. Here, we study bacterial evolution in antibiotic landscapes with a quantitative model where bacteria evolve under the stochastic processes of proliferation, death, mutation and migration. Numerical and analytical results show that cell motility can both accelerate and decelerate bacterial adaptation by affecting the degree of genotypic mixing and ecological competition. Moreover, we find that for sufficiently high rates, cell motility can limit bacterial survival, and we derive conditions for all these regimes. Similar patterns are observed in more complex scenarios, namely where bacteria can bias their motion in chemical gradients (chemotaxis) or switch between motility phenotypes either stochastically or in a density-dependent manner. Overall, our work reveals limits to bacterial adaptation in antibiotic landscapes that are set by cell motility.

Active motility is a defining feature of many cell types, governing their ecology and physiological functions. Immune cells such as neutrophils patrol multicellular organisms in their relentless search for invading microbes. Sperm cells actively swim and search ovules to fuse with. Growth cones of neurons seek their synaptic targets. Active motility is also fundamental for many unicellular organisms ranging from bacteria to amoeba and algae. It allows them to find nutrients, light or a host, and avoid toxic compounds, predators or parasites. In particular, bacterial motility has been thoroughly studied once it became understood as key for the reproductive success of bacteria and, more specifically, for their ability to cause disease[1–7]. Despite these important realizations, we know little about how cell motility contributes to bacterial adaptation, namely to the evolution of antibiotic resistance, which is a major public health concern[8].

Laboratory experiments suggest that bacterial motility is important for bacterial adaptation in antibiotic landscapes where cells can move through different concentrations of antibiotics[9–11]. In particular, it was found that in spatially heterogeneous environments, bacteria evolve antibiotic resistance faster than in homogeneous conditions[9]. However, cell motility was not controlled in these experiments, and its

precise contribution to the evolutionary dynamics found is not known. Notably, other authors did not find an accelerated adaptation in their antibiotic landscapes and argued that the discrepancy likely derived from the ability of bacteria to experience the diverse antibiotic concentrations, which was different in their experiments[10].

Arguably, the best understood link between cell motility and the ability of bacteria to cope with antibiotics comes from bacterial swarming, a form of group motility on surfaces where cells are particularly resilient to antibiotic stress[12,13]. More precisely, in swarming conditions, cell motility is thought to reduce exposure of individual cells to antibiotics, which leads to antibiotic tolerance[14]. While there are many studies relating bacterial swarming and phenotypic resistance[12,14–21], the effect of swarming motility on the evolution of genetic resistance has not been explored. In addition to swarming, other links between bacterial motility and antibiotic landscapes have been reported. In particular, it has been shown that antibiotics trigger motile responses[22–24]. However, the impact of such bacterial behaviour on the evolution of antibiotic resistance was not addressed. In short, one can find works that study bacterial evolution in antibiotic landscapes, but these do not explore the role of cell motility explicitly; and

[1]Department of Applied Mathematics and Theoretical Physics, Centre for Mathematical Sciences, University of Cambridge, Wilberforce Road, Cambridge CB3 0WA, UK. [2]Mathematical Institute, University of Oxford, Woodstock Road, Oxford OX2 6GG, UK. [3]Department of Veterinary Medicine, University of Cambridge, Madingley Road, Cambridge CB3 0ES, UK. ✉e-mail: n.m.oliveira@damtp.cam.ac.uk

one can find works that study bacterial motility upon antibiotic exposure, but these do not explore evolutionary timescales.

In addition to these experimental works, several mathematical models have been developed to understand how concentration gradients of antibiotics and other drugs affect resistance evolution[25–31], and a key conclusion is that antibiotic gradients accelerate the evolution of antibiotic resistance[25,26]. These quantitative models consider bacterial movement between environments with different antibiotic concentrations, such as different organs or parts of the human body, but the migration considered is essentially a passive and rare event. This assumption is in sharp contrast with the migration rates of bacteria evolving in natural and experimental settings, where cells can move at speeds up to ~ 50 μm/s, and the fact that in these environments bacteria can experience steep and stable gradients of antibiotics and other drugs[9–11,22].

In this work, we build upon these quantitative models and study how bacterial motility affects the evolution of antibiotic resistance in spatially heterogeneous environments with different

concentrations of antibiotics. We find that bacterial motility can govern the spatiotemporal dynamics of antibiotic resistance evolution.

## Results

### Model overview and basic dynamics

We are interested to understand how bacterial motility shapes the evolution of resistance in heterogeneous environments where cells experience spatial gradients of antibiotics. We build upon the so-called staircase model[25], which was developed to understand bacterial evolution in drug gradients. The staircase model is a lattice model, which considers cells of specified genotype $g \in \{1, ..., L\}$ (of increasing resistance) and spatial compartments $x \in \{1, ..., L\}$ (of increasing antibiotic concentration), Fig. 1a. An initial population of susceptible cells $g = 1$ at position $x = 1$ evolves via stochastic processes of death (rate $\delta$), movement to a neighbouring spatial compartment (rate $v$), mutation (rates $\mu_f/\mu_b$ for forward/backward), or division (maximal rate $r$)[25]. In this model, antibiotics inhibit the

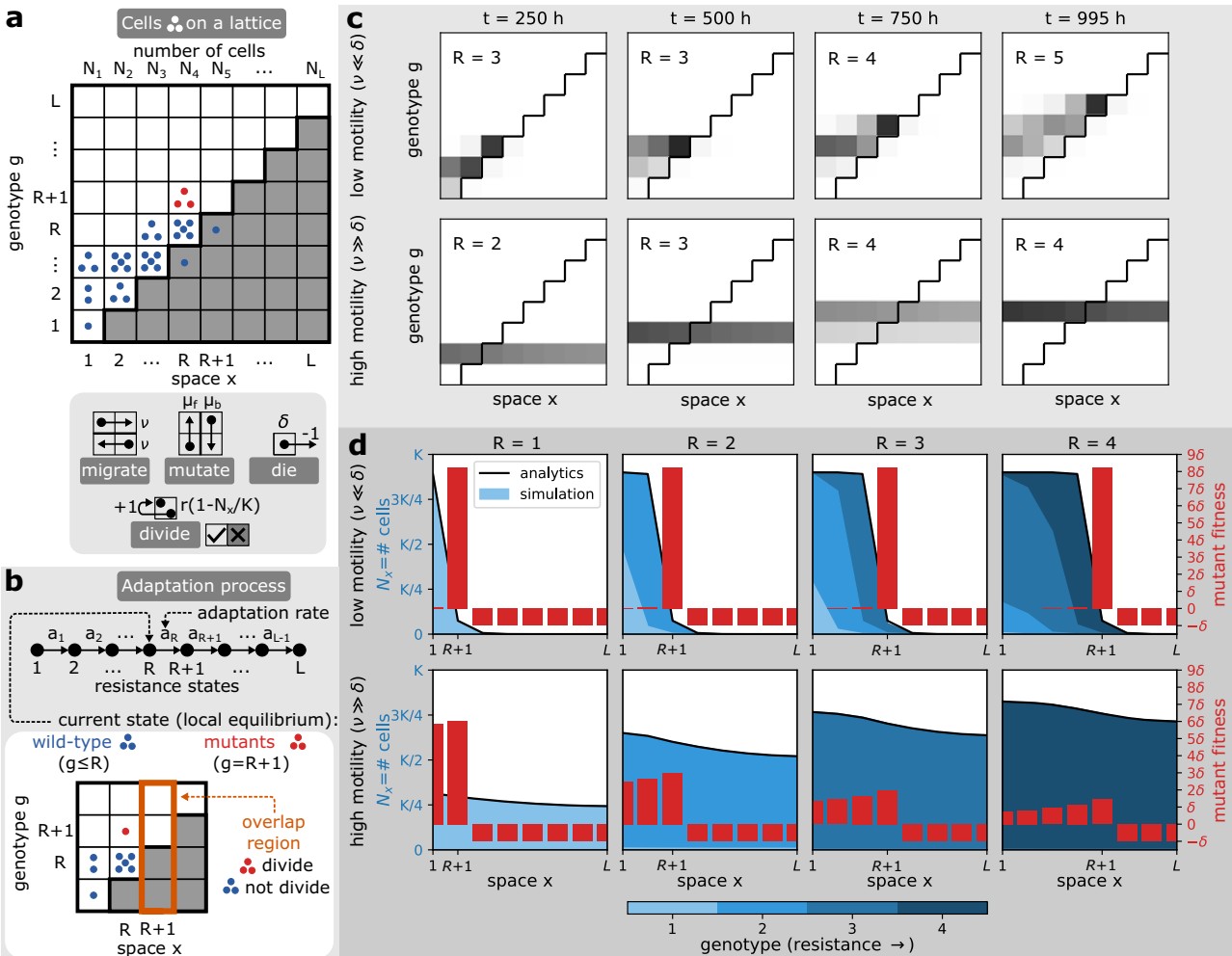

**Fig. 1 | Model overview and basic evolutionary dynamics. a** Model description. In the staircase model, bacterial cells (blue and red dots) have a specified position $x$ (of increasing antibiotic concentration) and a genotype $g$ (of increasing resistance). The population adapts to a stable spatial antibiotic gradient as cells migrate, mutate, die and divide at specified stochastic rates. Antibiotics prevent cell division of susceptible genotypes $g < x$ (shaded region under the staircase). **b** Adaptation process. Bacterial adaptation in the staircase model can be conceptualised as a series of random jumps between locally stable resistance states $R$ that happen at an adaptation rate $a_R$. At a given time, $R$ denotes the genotype of highest resistance in the population, as well as the highest spatial compartment where this genotype can

divide. Genotypes $g \leq R$ are defined as wild-type (blue dots) and genotype $g = R + 1$ as mutant (red dots). Importantly, in the overlap region $x = R + 1$, mutant cells can divide but wild-type cells cannot. **c** Snapshots of the population evolving antibiotic resistance for different motility rates, $v = 0.01/h$ (low motility) and $1/h$ (high motility). Shades of grey represent density of cells, where black represents highest density. The adaptive dynamics of bacteria with low (top) and high (bottom) motility is qualitatively different. **d** Population profiles and fitness along the antibiotic gradient. Different motility regimes affect the distribution of wild-type bacteria along the gradient (blue area), which shapes the fitness of resistant mutants (red bars). Parameters: $L = 8$, $K = 10^5$, $r = 1/h$, $\delta = 0.1/h$, $\mu_b = 10^{-4}/h$, $\mu_f = 10^{-7}/h$.

growth of susceptible genotypes $g < x$. Therefore, the diagonal of position-genotype lattice in Fig. 1a defines a staircase which separates areas where cells can divide (above the staircase, white) from areas where they cannot (below the staircase, grey). Moreover, the division rate of the $N_x$ cells at position $x$ is logistic with a carrying capacity $K$:

$$\begin{cases} r(1 - N_x/K), & \text{if } x \le g \text{ and } N_x \le K, \\ 0, & \text{otherwise}. \end{cases} \quad (1)$$

The processes of birth, death, mutation and migration drive bacterial adaptation up the antibiotic gradient. Assuming low mutation rates ($\mu_f \ll \delta, v, r$) and a large carrying capacity ($K \gg 1$)[25,26], this adaptation corresponds to random jumps $R \to R + 1$ between relatively stable population states $R \in \{1, \dots, L\}$, which we define as resistance states (Fig. 1b). We note, however, that $R$ can be understood as both the genotype $g$ of highest resistance in the population and the compartment of highest antibiotic concentration $x$ where this genotype can divide because the genotypic and spatial dimensions are interchangeable in this model. Genotypes $g \le R$ are defined as wild-type and genotype $g = R + 1$ as mutant. Resistance $R$ increases to $R + 1$ when mutants outcompete the wild-type in the overlap region at a position $x = R + 1$. The overlap region is the region where the population first adapts because it is the only spatial compartment where the regions where mutant cells can grow and regions where wild-type cells cannot grow overlap (Fig. 1b).

We start by studying how different motility rates $v$ affect the evolutionary dynamics of bacteria in this model as previous work[25,26] only considers situations where bacteria are unlikely to move between different antibiotic concentrations during their lifetime ($v < \delta$). In particular, we start by describing and comparing bacterial adaptation when cells rarely move during their lifespan ($v < \delta$) and situations where motility is common ($v > \delta$). Unexpectedly, our simulations show that bacterial adaptation is qualitatively different for low and high motility (Fig. 1c, Supplementary Movie 1). To understand this difference, we first study the qualitative properties of the stable population at a fixed resistance state $R$ and, in the next section, we focus on the adaptation rate $a_R$ at which the jump in resistance $R \to R + 1$ occurs (Fig. 1b).

Notably, we find that the low and high motility regimes differ in the shape of the wild-type population $N_x$ (SI Theorem 1), which impacts the mutant fitness landscape (Fig. 1d) that is defined by the net growth rate of mutant cells:

$$\begin{cases} r(1 - N_x/K) - \delta, & \text{if } x \le R + 1, \\ -\delta, & \text{otherwise}. \end{cases} \quad (2)$$

Moreover, we find that at low motility ($v < \delta$, Supplementary Movie 1), bacterial adaptation is limited by the movement of the first mutant into the overlap region, where mutant cells have low competition from wild-type cells and high fitness to set up a mutant population (Fig. 1d). In this case, bacterial adaptation is located at the population front (SI Theorem 3), while the rest of the population forms a typical inclined comet tail (Fig. 1c) corresponding to a diversity of strains with different antibiotic susceptibility (Fig. 1d). At high motility ($v > \delta$, Supplementary Movie 1), the wild-type is maintained even in regions where it cannot grow by influx from compartments where it can (SI Theorem 3), and, therefore, competes with mutants by contributing to the carrying capacity. The fitness of mutants is decreased in the overlap region and leveled across the region where mutants can divide (Fig. 1d). Therefore, mutant cells grow slowly and outcompete the wild-type everywhere. In these conditions, the population is made of a single strain that can grow in $x \le R$ (Fig. 1d) and the comet tail is now horizontal (Fig. 1c).

## Cell motility accelerates and decelerates bacterial adaptive evolution

We have shown that bacterial adaptation in antibiotic gradients is qualitatively different at low and high motility regimes. To gain quantitative understanding about this difference, we now study how motility affects the adaptation rate $a_R$ (Figs. 1b, 2a), which is defined by two waiting times:

1. The evolutionary time $T_{evo}^R$, defined as the earliest time when the wild-type produces a first mutant in the overlap region (i.e., mutants reach a founder state in Fig. 2a).
2. The ecological time $T_{eco}^R$, defined as the earliest time when the first mutant establishes a mutant population that becomes larger than the wild-type in the overlap region (i.e., mutants reach a winner state in Fig. 2a).

We follow Hermsen et al.[26] and define adaptation rate as the rate at which mutants reach consecutive founder states in Fig. 2a:

$$a_R = \frac{1}{\mathbb{E} T_{eco}^{R-1} + \mathbb{E} T_{evo}^R}, \quad (3)$$

where $\mathbb{E}$ denotes the expected value of the random waiting times. To determine the adaptation rates, we record the times of founder states in computer simulations (Fig. 2b and Methods), and compute the waiting times by a combination of analytical and numerical techniques that confirm and extend our simulations (Fig. 2b, 2c and Methods). Alternatively, one can define adaptation rate considering consecutive winner states instead of founder states (Supplementary Fig. 1a, b). However, as we note below, this alternative definition of adaptation rate does not affect our conclusions.

How does the adaptation rate $a_R$ vary in the two motility regimes we identified earlier? Our simulations show that in the low motility regime ($v < \delta$), increasing motility accelerates the adaptation rate of bacteria (Fig. 2b, Supplementary Fig. 1c), in accordance with previous theory[25,26]. Our analysis shows that in these conditions, the evolutionary time is much larger than the ecological time (Fig. 2c), because mutants have high fitness in the overlap region and can grow quickly (Fig. 2d). For this reason, Hermsen et al.[26] that considered low motility only, neglected the ecological time ($T_{eco}^R \approx 0$, Fig. 2a). Moreover, Hermsen et al.[26] noticed that the rate at which the population front advances becomes constant (i.e., it is independent of $R$) and implicitly identified all evolutionary times $T_{evo}^R = T_{evo}$. Our work supports this claim: the adaptation rate $a_R$ is identical for all $R$ when motility is low (Fig. 2b) and the shape of the population front is independent of $R$ (Fig. 1d, SI Theorem 3). Under these conditions, motility helps the first mutant to move from the population front into the overlap region and decreases the dominant waiting time, the evolutionary time $T_{evo}$. Therefore, in the low motility regime, there is a positive relationship between cell motility and bacterial adaptation (i.e., cell motility accelerates bacterial adaptation within this regime).

In contrast, at high motility ($v > \delta$), there is a negative correlation between cell motility and bacterial adaptation (i.e., cell motility decelerates bacterial adaptation within this regime), see Fig. 2b, Supplementary Fig. 1c. In this regime, the ecological time becomes larger than the evolutionary time (Fig. 2c) because mutant fitness is significantly decreased, which results from the increased competition for space with a large wild-type population in the overlap region. This competition is amplified with increasing motility $v$ and resistance state $R$, which promote the spread and size of the population (Fig. 2c).

The reduction in adaptation rate with higher motility becomes even more important if the wild-type has negative spatially-averaged fitness, which is defined as $\bar{f} = rR/L - \delta$ with $r$ being the birth rate, $\delta$ death rate, and $R/L$ the proportion of space where wild-type cells can divide. By definition, environments with this property, $\bar{f} < 0$, cannot sustain the wild-type if the system is spatially homogeneous, and we

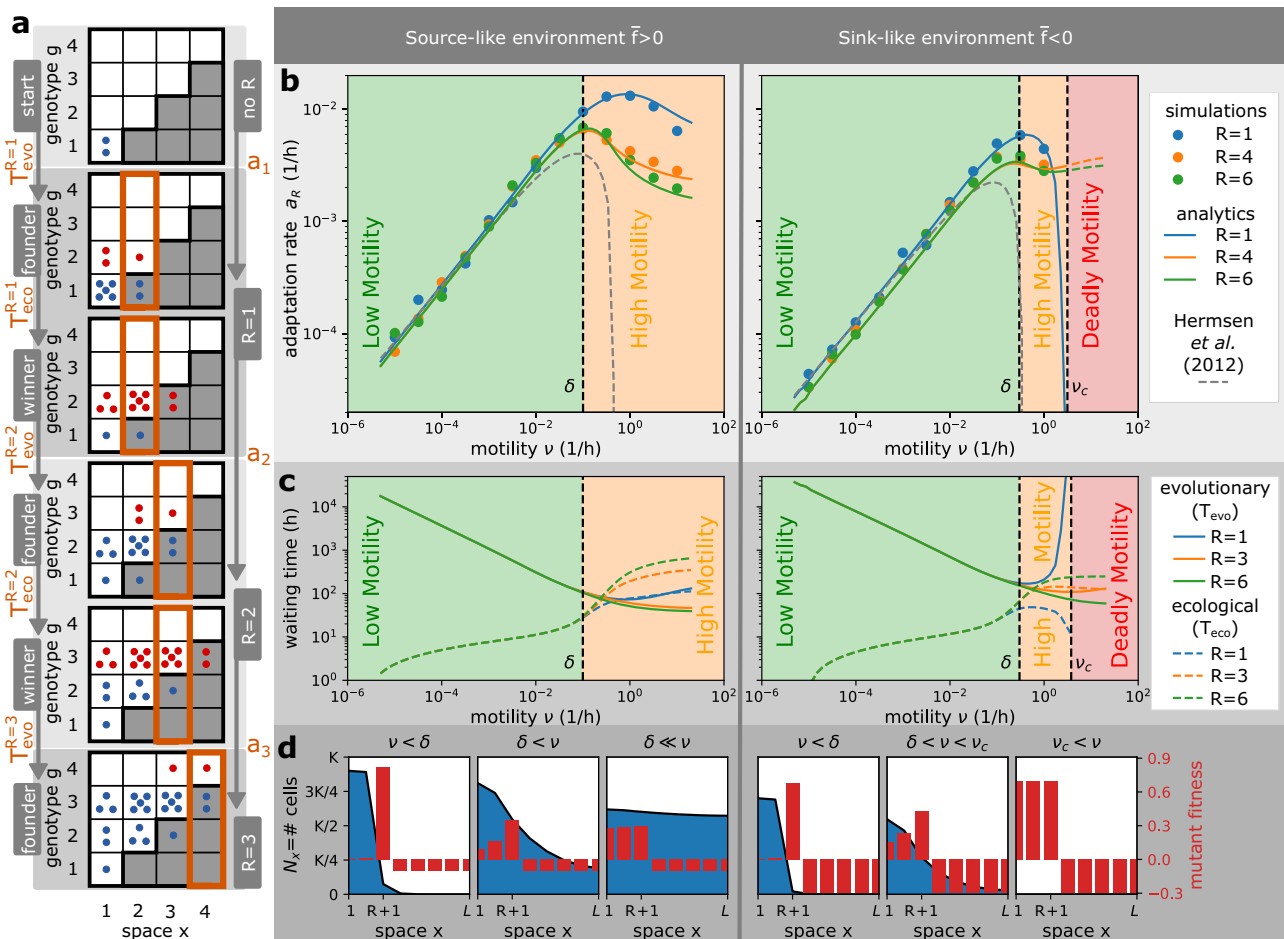

**Fig. 2 | Motility can accelerate and decelerate the rate of bacterial adaptation to antibiotics. a** Definitions. The adaptation process in the staircase model corresponds to jumps between resistance states $R \to R+1$ that happen at rate $a_R$. The resistance states $R$ are characterised by founder states (a first mutant appears in the overlap region) and winner states (mutants outgrow wild-type cells in the overlap region), which are separated by evolutionary $T_{evo}^R$ and ecological $T_{eco}^R$ waiting times. Following Hermsen et al.[26], we define the adaptation rate $a_R$ as the rate at which consecutive founder states appear, but we also consider an alternative definition based on the winner states and show it does not affect our conclusions (Supplementary Fig. 1). The evolutionary dynamics depends on the average wild-type fitness that is defined as $\bar{f} = rR/L - \delta$ with $r$ being the birth rate, $\delta$ death rate, and $R/L$ the proportion of space where wild-type cells can divide. **b** Adaptation rate as a function of motility rate. The lines represent analytical prediction and the symbols represent averages over 50 simulations, with errors smaller than the symbols. The adaptation rate increases with motility in the low motility regime, but decreases in the high motility regime, i.e., motility accelerates adaptation in the low motility regime, but decelerates adaptation in the high motility regime. If the wild-type has a negative average fitness and motility above a critical motility $v_c$, it cannot survive and adapt (deadly motility regime). The current resistance state $R$ does not affect the adaptation rate when motility is low, but decreases the adaptation rate when

motility is high. We plot the predicted dynamics from the analytical theory of Hermsen et al.[26] (dashed line), which matches our results for lower motility only, but note that this theory was developed under the assumption of low motility rates and the interpolation for higher motility shown for completeness is not valid. **c** Relative importance of evolutionary and ecological times. At low motility, bacterial adaptation is largely dependent on the evolutionary time $T_{evo}^R$. At high motility, however, the ecological time $T_{eco}^R$ becomes equally important due to increased competition between strains. Near critical motility $v_c$, bacterial adaptation is again mainly limited by the evolutionary time $T_{evo}^R$. At high motility, the number of compartments where the wild-type can divide $R$ increases the ecological time $T_{eco}^R$ and decreases the evolutionary time $T_{evo}^R$. **d** The importance of wild-type profiles. At low motility regimes, motility promotes the movement of mutants from the population front into the overlap region ($T_{evo}^R$ reduced), where mutants have larger fitness than the wild-type and grow quickly ($T_{evo}^R \gg T_{eco}^R$). At high motility regimes, motility promotes the spatial spread of the wild-type population with genotypes $g \le R$, so that the first mutant is produced quickly but mutant fitness is decreased due to higher competition for space with the wild-type ($T_{evo}^R < T_{eco}^R$). Parameters: $L = 8$, $K = 10^5$, $r = 1/h$, $\delta = 0.1/h$ for source-like environment, $\delta = 0.3/h$ for sink-like environment, $\mu_b = 10^{-4}/h$, $\mu_f = 10^{-7}/h$.

call them sink-like (as opposed to source-like with $\bar{f} > 0$). This terminology is inspired by the source-sink theory, which defines a source (resp. sink) as an environment with positive (resp. negative) local fitness[32], as opposed to the average fitness in our model that considers fitness across all compartments. Notably, we find that cell motility creates sufficient mixing in sink-like environments above a critical motility threshold ($v > v_c$), and that in these conditions the wild-type cannot survive and adapt (Fig. 2b, SI Theorem 1, SI Theorem 4, SI Corollary 1). Therefore, we call it a deadly motility regime. We note that the deadly motility regime would not occur if resistant cells were present in the population from the beginning, which would increase

the average wild-type fitness from $\bar{f} < 0$ to $\bar{f} > 0$ due to higher $R$. In the latter case, the same environment would become source-like, with survival and adaptation possible for any motility (dashed lines in Fig. 2b). We further note that the existence of the deadly motility regime requires the system to be closed, with no external import of wild-type cells.

In summary, our analysis identifies three distinct regimes of bacterial adaptation in antibiotic gradients that depend on the motility rate of cells, which we call low motility, high motility and deadly motility regimes (Fig. 2b). Notably, we find the same regimes when an alternative definition of adaptation rate is considered (Supplementary

Fig. 1c), when antibiotic resistance carries fitness costs (Supplementary Fig. 2, Supplementary Note 4), when mutation rates are high (Supplementary Fig. 3, Supplementary Note 4), when non-growing cells compete with growing cells at a reduced rate (Supplementary Fig. 4, Supplementary Note 4), and when instead of bacteriostatic antibiotics we consider bactericidal antibiotics (Supplementary Fig. 5, Supplementary Note 4). Unexpectedly, we find the same regimes even when cell motility is biased (Supplementary Fig. 6, Supplementary Note 4, Supplementary Movie 2). It has recently been shown that antibiotics can trigger both positive[22] and negative[24] chemotaxis in bacterial cells, but our results show that such biased motility does affect our key findings (Supplementary Fig. 6). Overall, these different scenarios highlight the robustness of our conclusions.

## Evolutionary dynamics of phenotypically heterogeneous populations

So far, we have considered evolving populations that are phenotypically homogenous and bacterial cells that always move at the same rate. The same simplifying assumption has been made in previous models of bacterial evolution in antibiotic gradients[25–31]. However, in reality, bacteria can change their motility rate, and their communities are phenotypically heterogeneous in terms of cell motility. For example, one of the most important transitions in a bacterial life-cycle, from free-living (plankton) to surface-attached communities (biofilms), is associated with major changes in cell motility. Planktonic bacteria can move at $45\,\mu m/s^2$, while biofilm bacteria can move at $1\,\mu m/min$ or less[33]. The plankton-biofilm switch can occur both stochastically[34] or as a response to multiple factors, such as cell density[35] or the presence of other strains[36]. Another well-known transition in bacterial populations that is associated with changes in cell movement is the swarming phenotype, where bacteria move collectively on semi-solid surfaces when a threshold of cell density is reached[37]. We found it important to understand if this phenotypic heterogeneity could affect the evolutionary dynamics we presented earlier.

To address this issue, we now consider populations harbouring two motility phenotypes where cells can switch between phenotypes either stochastically (Fig. 3) or as a response to local cell density (Fig. 4). In practice, for stochastic switching, we add a new dimension

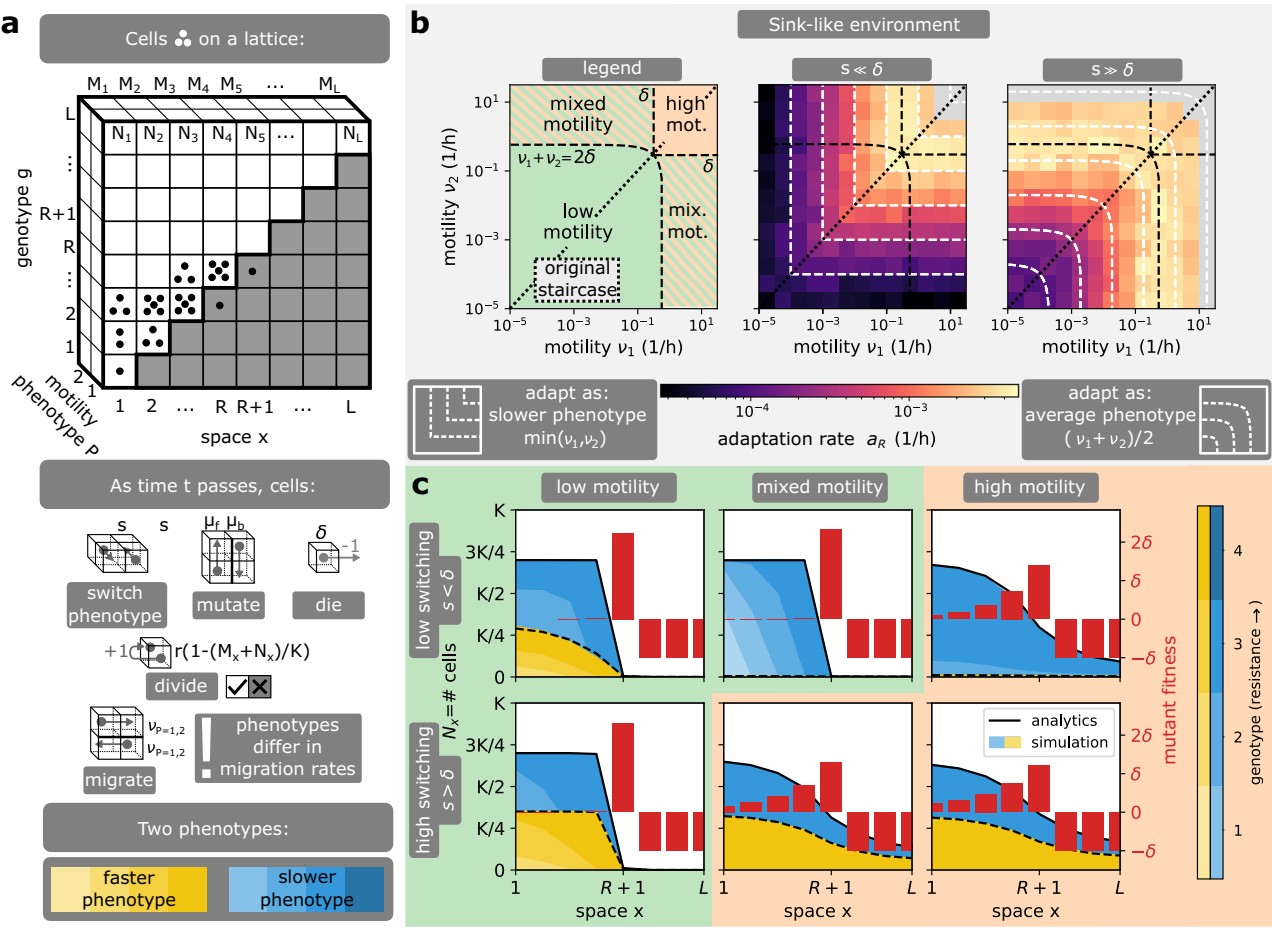

**Fig. 3 | Effect of switching between motility phenotypes stochastically. a** Model description. Phenotypes of motility $v_{1,2}$ are added as an extra dimension to the staircase model, with individual cells allowed to switch between these phenotypes at rate $s$. **b** Adaptation rate heatmap on the $(v_1, v_2)$ plane for different switching rates $s$. The $(v_1, v_2)$ plane can be partitioned into different combinations of adaptation regimes and its diagonal corresponds to a population of a single motility. In this model, bacteria adapt as if all cells had the same effective motility, which corresponds to the intersection of the level sets (white dashed lines) with the diagonal. At low switching rate ($s = 0/h$), the effective motility matches the slower motility present in the population $\min(v_1, v_2)$. At high switching rate ($s = 5/h$), the effective motility matches the average motility $(v_1 + v_2)/2$. When the environment is sink-like, a deadly motility regime (grey) exists in the high motility combination, and also appears in the mixed motility combination when the switching rate is high. **c** Wild-type profiles for different switching rates $s$. At low switching rate ($s = 10^{-3}/h$), the wild-type profile is dominated by the slower phenotype. At high switching rate ($s = 5/h$), the wild-type profile coincides with the profile of a single average-motility phenotype. Therefore, the adaptation in low (resp. high) motility combinations follows the low (resp. high) motility regime irrespective of switching rate $s$, but a change of $s$ in the mixed motility combination can change the adaptation regime. In short, stochastic motility switching shapes the evolution of antibiotic resistance by determining the effective motility of bacterial populations. Parameters: $L = 8$, $K = 10^5$, $r = 1/h$, $\delta = 0.3/h$, $\mu_b = 10^{-4}/h$, $\mu_f = 10^{-7}/h$.

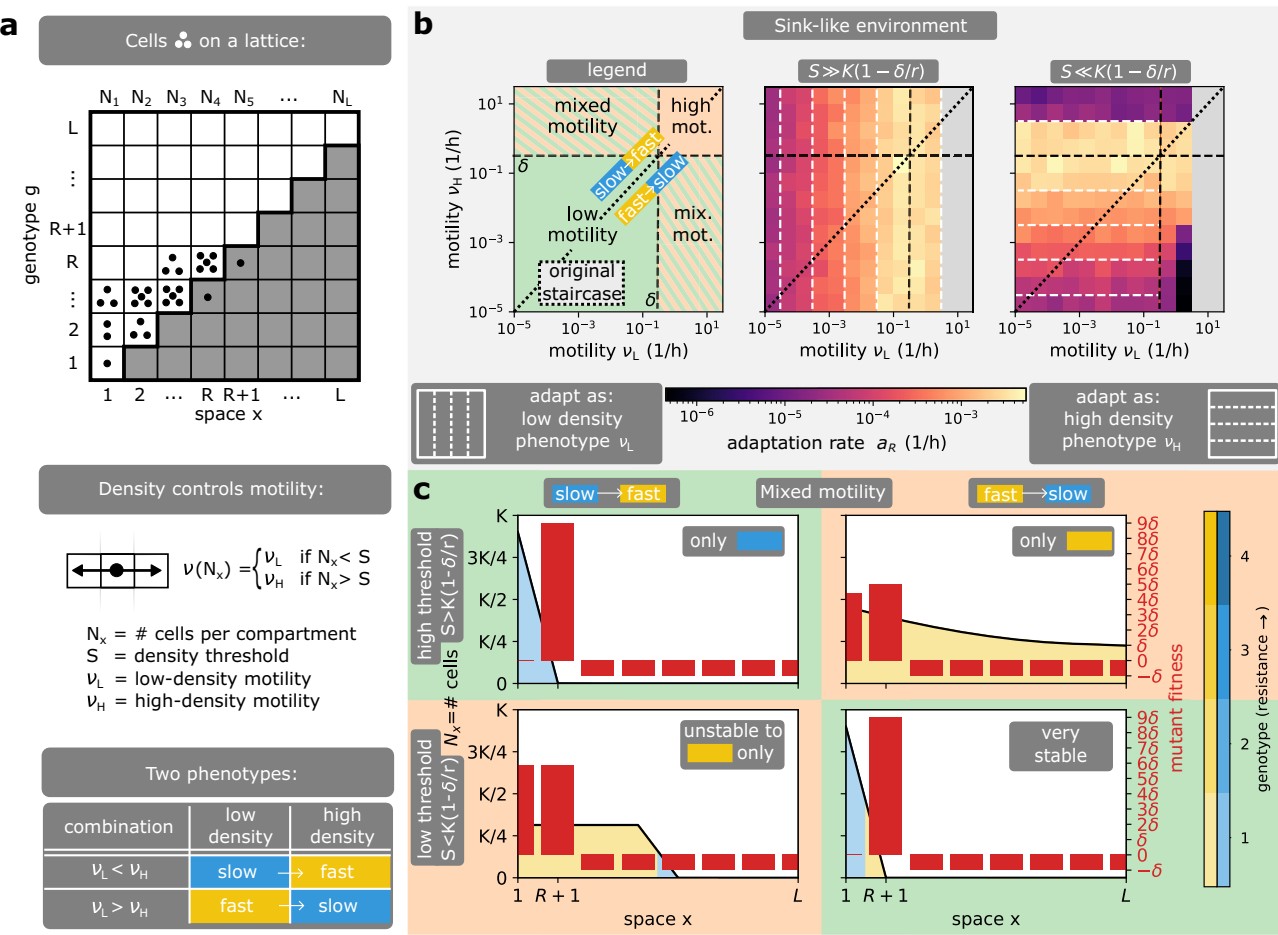

**Fig. 4 | Effect of density-dependent motility. a** Model description. In this model, bacterial motility $v(N_x)$ depends on the local number of cells $N_x$. If the cell density $N_x$ at position $x$ is below (resp. above) the switching threshold $S$, all cells at position $x$ move at low-density (resp. high-density) motility $v_L$ ($v_H$). **b** Adaptation rate heatmap on the $(v_L, v_H)$ plane for different switching thresholds $S$. The $(v_L, v_H)$ plane can be partitioned into different combinations of adaptation regimes and its diagonal corresponds to a population of a single motility. The diagonal separates slow-to-fast and fast-to-slow switching combinations. Bacteria generically adapt as if all cells had the same effective motility, which corresponds to the intersection of the level sets (white dashed lines) with the diagonal. This effective motility generically matches the low-density (resp. high-density) motility at high (resp. low) threshold $S = 9.5 \times 10^4$ (resp. $S = 100$). At low threshold $S$, this generic rule has three exceptions. First, the adaptation rate is reduced in the mixed motility combination of fast-to-slow switching. Second, the adaptation rate is reduced for slow-to-fast switching when the environment is sink-like and the high-density motility

phenotype moves above the critical motility. Third, a deadly motility regime (grey) can only occur if the initial population is of low density and the low-density motility phenotype moves above the critical motility. **c** Wild-type profiles in the mixed motility combination. Wild-type profiles match the low-density motility phenotype at high threshold $S = 9.5 \times 10^4$. At low threshold $S = 10^{4.5}$, the low-density motility phenotype appears only at the front while the bulk of the population is in the high-density motility phenotype. This composition of phenotypes can reduce the antibiotic exposure of the bacterial population: in slow-to-fast switching, the spatial expansion of the fast (yellow) high-density motility phenotype is reduced in sink-like environments, and in fast-to-slow switching, a very slow (blue) and stable population occupies low antibiotic concentrations. In short, density-dependent motility shapes bacterial adaptation by determining the effective motility of bacterial populations and affecting their antibiotic exposure. Parameters: $L = 8$, $K = 10^5$, $r = 1/h$, $\delta = 0.3/h$, $\mu_b = 10^{-4}/h$, $\mu_f = 10^{-7}/h$.

of motility phenotypes to our position-genotype lattice and allow cells to switch between phenotypes at a stochastic rate $s$, where each phenotype is characterized by different motility rates, $v_1$ and $v_2$ (Fig. 3a). Density-dependent motility $v(N_x)$ is modelled differently, and the switch between motility phenotypes is modelled by a step-function that is controlled by the local number of cells $N_x$ and a switching density threshold $S$ (Fig. 4a). If cell density $N_x$ at position $x$ is below (resp. above) the switching threshold $S$, all cells at position $x$ move at low-density (resp. high-density) motility $v_L$ (resp. $v_H$). Density-dependent motility can also be modelled explicitly (Supplementary Note 6), and we note below that it does not affect our conclusions.

Interestingly, we find that in both models, phenotypically heterogeneous populations of bacteria generically behave as if all cells had a single effective motility phenotype whose motility is controlled by the switching rate $s$ in stochastic switching, or the switching

threshold $S$ if switching is density-dependent. Therefore, the previously described adaptation regimes exist in these models and are governed by an effective motility that we characterize next. For stochastic switching, the effective motility matches the motility of the slower phenotype $v_- = \min(v_1, v_2)$ when the switching rate is low ($s \ll \delta$, Supplementary Movie 3) and it matches the average motility $v_+ = (v_1 + v_2)/2$ when the switching rate is high ($s \gg \delta$, Supplementary Movie 3), see Fig. 3b and SI Theorems 6, 7, 8 and SI Corollary 2, 3. This result stems from the fact that the slower phenotype is naturally selected for in the absence of phenotypic switching, given that the slower phenotype spends more time in regions of low antibiotic concentration where it can proliferate, and from the fact that large stochastic switching quickly equilibrates the abundance of both phenotypes (Fig. 3c). For density-dependent motility, the effective motility matches the low-density motility phenotype $v_- = v_L$ at high switching threshold ($S > K(1 - \delta/r)$) and the high-density motility

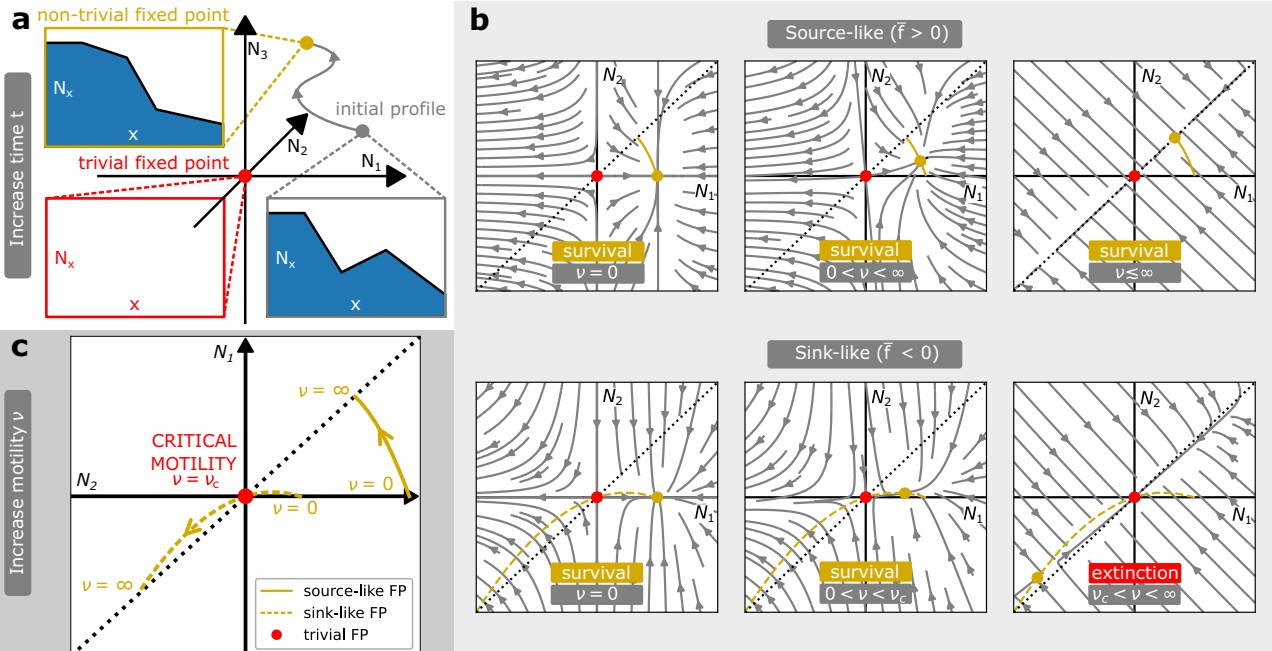

**Fig. 5 | Cell motility as a factor that limits bacterial survival in sink-like environments. a** Formation of the wild-type profile. The formation of a wild-type profile corresponding to a fixed resistance state $R$ is described by mean-field equations, which explain the flow of the initial profile in the profile space of cell numbers $(N_1, ..., N_L)$. This flow admits two fixed points: a trivial fixed point $N_x = 0$ (red, population extinction if stable) and a non-trivial fixed-point (yellow, population survival if stable). **b** Phase portraits of the wild-type profile formation in the staircase model with $L = 2$, $R = 1$ (the so-called source-sink model[25]). Phase portraits are shown for different motility rates $v$ and different environment types (source-like/sink-like), depending on the average wild-type fitness $\bar{f} = r/2 - \delta$. Only two topologically distinct types of flow are possible: the non-trivial fixed point globally attracts all possible wild-type profiles with $N_x > 0$ (population survival), or the trivial fixed point globally attracts all possible wild-type profiles with $N_x > 0$ (population extinction). Extinction occurs in sink-like environments ($\bar{f} < 0$) with motility above the critical motility $v > v_c$. **c** Critical motility $v_c$ corresponds to a bifurcation of this dynamical system. When motility $v$ is varied, the non-trivial fixed point moves through the space of possible profiles. At low motility, the non-trivial fixed point is stable and corresponds to a wild-type profile that predominantly occupies spatial positions $x \leq R$ where it can divide. As motility increases, the wild-type profile gets increasingly levelled across all spatial compartments, and its dynamics becomes governed by the average fitness $\bar{f}$. Precisely when the environment is sink-like $\bar{f} < 0$, the stable non-trivial fixed point (surviving wild-type) collides with the unstable trivial fixed point (extinct wild-type), and they exchange their stability at the critical motility $v_c$. In short, highly motile populations experience an average environment, which can drive their extinction if the environment is sink-like. Parameters: $L = 8$, $K = 10^5$, $r = 1/h$, $\delta = 0.75/h$ for source-like environment, $\delta = 0.25/h$ for sink-like environment.

phenotype $v_+ = v_H$ at low switching thresholds ($S < K(1 - \delta/r)$, Fig. 4b). This result stems from the fact that the population cannot reach sufficient density for the high-density motility phenotype when the threshold is high, while a lower threshold switches the population bulk into the high-density motility phenotype.

Accordingly, if the population harbours phenotypic heterogeneity of the same motility regime, the adaptation regime corresponding to the effective motility is not changed by the stochastic rate $s$ and the density threshold $S$. However, for populations that harbour motility phenotypes from different motility regimes (mixed motility in Figs. 3b and 4b), the level of switching shapes the adaptation regime of bacteria (Figs. 3 and 4c). For stochastic switching, such heterogeneous populations have an effective motility in the low motility regime at low switching ($v_- < \delta$) and in the high (resp. deadly) motility regime at high switching ($v_+ > \delta$, resp. $v_+ > v_c$). Put differently, a gradual increase in switching rate changes the adaptation regime between the low, high and deadly motility regimes (Fig. 3c, Supplementary Fig. 7a, SI Theorems 6, 8). Similarly to phenotypically homogeneous populations, the deadly motility regime can only occur if the average wild-type fitness is negative (sink-like environment, Fig. 3b, Supplementary Fig. 7c, SI Theorem 8, SI Corollary 3).

For highly heterogenous populations with density-dependent motility, the gradual change in switching threshold $S$ also dictates the effective motility and the corresponding adaptation regime, but there are three exceptions when the density threshold $S$ is low (Fig. 4b). First, when cells switch from fast to slow above the

threshold $S$, the population becomes very stable as the fast cells at the front quickly return to the slow population bulk (Fig. 4c). As a result, the population has decreased antibiotic exposure and adaptation rate (Fig. 4b). Second, a deadly motility regime occurs if the low-density motility (not the effective motility) is above the critical motility (Fig. 4b). Third, if the high-density motility is above the critical motility, the population does not go extinct but instead decreases its range and antibiotic exposure (Fig. 4c, Supplementary Movie 4), leading to a decreased adaptation rate (Fig. 4b). As the last two exceptions are related to critical motility, they are only important in sink-like environments where the wild-type has negative average fitness (Supplementary Fig. 8b, c). Similar conclusions are found when density-dependent motility is modelled explicitly (Supplementary Fig. 8).

With the exception of the highly heterogeneous populations with density-dependent motility, we showed that bacterial populations with different motility phenotypes evolve antibiotic resistance as if all cells had the same effective motility. Therefore, in these conditions, bacterial adaptation is characterised by the same adaptation regimes of phenotypically homogeneous populations. Switching between motility phenotypes can shape the adaptation regime under which bacteria evolve only if the motility of the different phenotypes is very different. But even in these cases, we find that our general conclusions hold, low effective motility accelerates bacterial adaptation while high effective motility decelerates bacterial adaptation. For very high effective motility, bacterial populations can perish.

**Cell motility as a factor that limits bacterial survival in sink-like environments**

We have seen how bacterial motility shapes the evolution of antibiotic resistance in a variety of models. We have shown that wild-type profiles form the basis of bacterial adaptation by creating mutants and dictating their fitness in antibiotic landscapes (Fig. 1d). Moreover, all our models admit a deadly motility regime, defined by a critical motility (Fig. 2b, Fig. 3b, Fig. 4b). Above the critical motility, the wild-type goes extinct and adaptation is not possible (Supplementary Movie 5). The goal of this last section is to explain the emergence of such critical motility regime, and to bring together the previous sections.

We start by studying analytically how the wild-type profile forms at a fixed resistance state $R$ (Fig. 1b, d). A wild-type profile $(N_1, ..., N_L)$ can be conceptualised as a point in the profile space of cell numbers (Fig. 5a) and its time-evolution can be described by the mean-field equations that are derived from the stochastic dynamics. We illustrate the general approach (SI Theorems 2, 3) using the staircase model with only two compartments ($L = 2, R = 1$), which is known as the source-sink model[25]. In this model, the mean-field equations for the wild-type profile formation are the following,

$$\dot{N}_1 = r(1 - N_1/K)N_1 - (\delta + \nu)N_1 + \nu N_2,$$
$$\dot{N}_2 = -(\delta + \nu)N_2 + \nu N_1. \tag{4}$$

This system can be analysed with dynamical systems theory, which studies qualitative features of systems of ordinary differential equations (ODE). More precisely, the time-evolution of this system on $\mathbb{R}^2$ can be characterised by phase portraits (Fig. 5b), which in this case have two fixed-points: a trivial fixed point $N_1 = N_2 = 0$ where the population goes extinct, and a non-trivial fixed point where the population survives, which is defined as

$$N_1 = K\left(1 - \frac{\delta(\delta + 2\nu)}{r(\delta + \nu)}\right),$$
$$N_2 = \frac{\nu}{\nu + \delta}N_1. \tag{5}$$

These phase portraits admit only two distinct types of trajectory flows (Fig. 5b). Either the non-trivial fixed point has positive physical densities $N_x > 0$ and globally attracts all possible wild-type profiles with $N_x > 0$ (i.e., the bacterial population survives); or the non-trivial fixed point is non-physical $N_x < 0$ and the trivial fixed point attracts all possible wild-type profiles with $N_x > 0$ (i.e., the bacterial population does not survive). These two types of phase portraits appear at different motility rates $\nu$. Furthermore, we can understand this dynamics with bifurcation theory, and study how the non-trivial fixed point moves through the profile space as motility $\nu$ increases from zero (Fig. 5c). At low motility, the non-trivial fixed point is stable and located in the non-negative quadrant $N_{1,2} \geq 0$. As motility increases, the dynamics is different in source-like ($\bar{f} > 0$) and sink-like ($\bar{f} < 0$) environments, where the average wild-type fitness of the source-sink model is $\bar{f} = r/2 - \delta$. If the environment is source-like, the non-trivial fixed point never leaves the quadrant $N_{1,2} > 0$ and remains stable. However, in sink-like environment, the non-trivial fixed point collides with the trivial fixed point through a transcritical bifurcation, which exchanges their stability. In these cases, there is a critical motility,

$$\nu_c = \frac{\delta(r - \delta)}{2\delta - r} > 0 \tag{6}$$

which corresponds to this bifurcation. The emergence of a bifurcation is necessary, since (a) the trivial fixed point must be stable as $\nu \to \infty$ when cells experience an average sink-like environment, and (b) the non-trivial fixed point must be stable as $\nu \to 0$ when cells grow locally in the source at $x = 1$ (SI Theorem 3). Therefore, these fixed points must

collide and exchange their stability through a transcritical bifurcation. This bifurcation theory argument is general and provides a robust mechanism for the emergence of the critical motility regime in all our models (Supplementary Note 2).

In short, and put differently in a more biological realm, a bacterial population can be driven to extinction by increasing the rate at which cells move in sink-like environments (i.e., in spatially heterogeneous environments where the average wild-type fitness is negative) because highly motile cells experience an average environment. This idea suggests that one can limit bacterial adaptation in heterogeneous landscapes by targeting cell motility.

## Discussion

In this work, we have used mathematical analysis to study how bacterial motility affects the evolution of antibiotic resistance in spatially heterogeneous environments that have different antibiotic concentrations. By doing so, our study contributes to the large body of literature that has recognized spatial heterogeneity as an important factor for bacterial adaptive evolution but where active motility of bacteria remains poorly explored[9–11,25–31].

We have identified three regimes of bacterial adaptation, which are defined by the degree of cell motility in evolving populations. Accordingly, we called them low motility, high motility and deadly motility regimes (Fig. 2b). The theory of the low motility regime has already been discussed[25,26], and has been compared with experimental works that studied how antibiotic gradients affect bacterial adaptation rate[9,10]. Interestingly, while Zhang et al.[9] reported an increased adaptation rate in their antibiotic gradients compared to well-mixed conditions, Baym et al.[10] did not find such an acceleration in theirs. We note that while the same bacterial species were used, the antibiotic landscape that cells experienced was very different in these two experimental systems, which would be sufficient to affect the adaptation rates according to our work (Fig. 2b). Such difference can emerge when cells move at different rates (Fig. 2b), or for different gradient steepness[27,28,31]. Indeed, these effects are interchangeable because increasing gradient steepness brings spatial points of fixed antibiotic concentrations closer, which resembles the effect of increasing effective motility.

This effect then suggests that the adaptation regime of an evolving population is governed by the antibiotic variability explored by bacteria during their lifetime. The adaptation rate can therefore be quantified by a dimensionless visiting number $V$, defined as the number of regions that differ in antibiotic concentrations and that are visited by an average cell during its life-time (Supplementary Note 7). In our model, $V = \nu/\delta$ as cells change antibiotic regions at a rate $\nu$ and live for time $1/\delta$. Therefore, $V < 1$ (resp. $V > 1$) corresponds to the low (resp. high) motility regime (Fig. 2). In experiments, the visiting number can be calculated from the characteristic cell speed $v$, doubling time in the absence of antibiotics $t$ and length-scale over which the drug concentrations vary on MIC scales $l$. The visiting number is $V = vt/l$ as cells explore length-scale of $vt$ during their lifespan, while antibiotics differ at length-scale $l$. We estimated the visiting numbers in Zhang et al.[9] and Baym et al.[10] and predict that cells in Baym et al.[10] were evolving in a low motility regime ($V \approx 0.1846 < 1$) and cells in Zhang et al.[9] in the high motility regime ($V \approx 120 > 1$). Notably, Baym et al.[10] observed adaptation at the population front and coexistence of resistance strains, which is consistent with a low motility regime (Fig. 1d), while Zhang et al.[9] observed that resistant mutants quickly invade the entire environment, consistent with high motility regime (Fig. 1d). If bacteria were evolving in different motility regimes as predicted by our visiting number, it may help to explain why the authors observed different rates of bacterial adaptation in their studies (Supplementary Table 1).

As a corollary of our analysis, we find that there is an optimal level of cell motility for bacterial adaptation in spatially heterogeneous

environments (Fig. 2b). A similar optimum has been identified in models of HIV[38] and cancer[39] drug resistance evolution. These models are analogous to our setting in that they consider the movement of viruses or cancer cells between compartments of different drug concentrations. However, the deadly motility regime was not identified in these works. It is not clear if these models do not allow for a deadly motility regime or if instead the authors simply did not explore their model fully. Since our models rest on general features of living cells, they may be useful to understand the evolutionary dynamics of cell types other than bacteria, namely cancer cells during chemotherapy, where the role of cell motility and drug gradients remains poorly explored[40,41].

Some of our findings are closely related to those from ecology works that study biological adaptation at range edges where species expand their range in an environmental gradient by dispersal and mutation[42–45]. In this literature, local adaptation is known to be prevented by high motility, the so-called motility load[43], where less fit wild-type genotypes migrate enough to increase competition with mutant genotypes, which decreases selection for the latter. As a result, genetic diversity is lower in their high motility regime when compared to the low motility regime, as we find in our work (Fig. 1d). This loss of genetic diversity in the ecological models is often associated with critical motility[43,46,47], which can be compared with our critical motility. Notably, the emergence of a critical motility in these works can be explained by the same bifurcation mechanism that we identified (Supplementary Note 2), which highlights the robustness of our mechanism across modelling frameworks. Moreover, due to its simplicity, the staircase model can be used to provide analytical insights into biological adaptation at range edges as in Lenormand[43], Nagylaki[48] and Kirkpatrick & Barton[42], while considering the important stochasticity of natural processes as in Polechova & Barton[45].

Our models make multiple simplifying assumptions regarding the biology of bacteria. While we extended our initial staircase model to account for some of this biology, such as the effect of resistance costs (Supplementary Fig. 2), level of mutation rates (Supplementary Fig. 3), differential competition between growing and non-growing cells (Supplementary Fig. 4), bactericidal antibiotics (Supplementary Fig. 5) and biased motility (Supplementary Fig. 6), these processes could be modelled differently. For example, we modelled resistance costs as a reduction in division rate (Supplementary Fig. 2), while one could study resistance costs that decrease motility rate. Moreover, we assumed implicitly that bacterial adaptation results from the vertical transmission of resistance genes only, and horizontal gene transfer (HGT) is known to affect the evolution of antibiotic resistance[49]. However, the staircase model is a closed system and it is known that for these systems, HGT does not have an important impact[50–54]. While considering the effect of immigration of cells was beyond the scope of this work, we tested the predictions for closed systems by implementing HGT in the staircase model and found that HGT does not affect our conclusions (Supplementary Fig. 9, Supplementary Note 4).

In addition to considering the effect of HGT more thoroughly, one may extend our models to account for other relevant biology of bacteria such as the existence of persister phenotypes, which can affect the stability of microbial communities[55], or the effect of phage infection, which can affect bacterial motility[24]. Regardless, our current results lead us to conclude that cell motility limits bacterial adaptive evolution in spatially heterogeneous environments. This realization then suggests that manipulating motility may prove to be a useful approach for controlling bacterial systems and their impacts on us.

## Methods
### Simulations
A major component of our methods is computer simulations of the staircase model and its extensions. Hereby, we describe how to simulate the basic staircase model. The staircase model treats bacterial

evolution in drug gradients as a continuous-time Markov process, which is a stochastic process whose future state depends only on the present state and not on the past states. A state of the staircase model is specified by the number of bacterial cells $N_{x,g}$ at each lattice point $(x, g) \in \{1, ..., L\}^2$ of the position-genotype lattice. The initial state of the staircase model at time $t = 0$ is chosen as $N_{1,1} = 10^{-3}K$ and $N_{x,g} = 0$ for other lattice points $(x, g) \neq (1, 1)$. The time-evolution of the initial state of the staircase model $N_{x,g}$ is determined by the following processes $i \in I$:

- death of a cell of genotype $g$ at position $x$ ($N_{x,g} \rightarrow N_{x,g} - 1$): rate $b_i = \delta N_{x,g}$,
- division of a cell of genotype $g$ at position $x \leq g$ when $\sum_g N_{x,g} \leq K$ ($N_{x,g} \rightarrow N_{x,g} + 1$): rate $b_i = r(1 - \sum_g N_{x,g})$,
- movement of a cell of genotype $g$ from position $x$ to position $x + 1$ ($N_{x,g} \rightarrow N_{x,g} - 1, N_{x+1,g} \rightarrow N_{x+1,g} + 1$): rate $b_i = \nu N_{x,g}$,
- movement of a cell of genotype $g$ from position $x$ to position $x - 1$ ($N_{x,g} \rightarrow N_{x,g} - 1, N_{x-1,g} \rightarrow N_{x-1,g} + 1$): rate $b_i = \nu N_{x,g}$,
- mutation of a cell at position $x$ from genotype $g$ to genotype $g + 1$ ($N_{x,g} \rightarrow N_{x,g} - 1, N_{x,g+1} \rightarrow N_{x,g+1} + 1$): rate $b_i = \mu_f N_{x,g}$,
- mutation of a cell at position $x$ from genotype $g$ to genotype $g - 1$ ($N_{x,g} \rightarrow N_{x,g} - 1, N_{x,g-1} \rightarrow N_{x,g-1} + 1$): rate $b_i = \mu_g N_{x,g}$. To simulate this time-evolution, we use the exact sampling of stochastic trajectories of the staircase model via a computationally efficient Next Reaction Method[56], which samples the next process that takes place and the time when it happens. Effectively, this method combines computationally efficient data structures and the Gillespie's algorithm[57]:

1. sample the next process $i$, which happens with probability $b_i/\sum_{j \in I} b_j$,
2. sample the waiting time $\Delta t$, which has distribution $\text{Exp}(\sum_{i \in I} b_i)$,
3. update the time $t \rightarrow t + \Delta t$ when the next process $i$ happens and execute the process $i$.

Using this sampling technique, the times that correspond to the founder and winner states can be measured for all resistance states $R$ (Fig. 2a). Founder states are defined by the first time when a single mutant cell ($g = R + 1$) appears in the overlap region ($x = R + 1$), while the winner states are defined by the first time when there is more mutant cells ($g = R + 1$) than wild-type cells ($g \leq R$) in the overlap region ($x = R + 1$). As in Hermsen et al.[26], we record the times that correspond to the founder states $T_i^R$ in $n = 50$ independent simulations $i$ and set $T_i^0 = 0$. The adaptation rate is estimated as,

$$a_R = \frac{n}{\sum_i (T_i^R - T_i^{R-1})}. \tag{7}$$

This expression matches the definition of the adaptation rate precisely when $n \rightarrow \infty$, because $\sum_i (T_i^R - T_i^{R-1})/n \rightarrow \mathbb{E}T_{\text{evo}}^R + \mathbb{E}T_{\text{eco}}^{R-1}$ by the law of large numbers. The standard deviations associated with taking finite $n = 50$ can be computed using the central limit theorem and we checked that their corresponding error bars are smaller than the symbols in Fig. 2b. As we use $n = 50$ simulations in all Figures and all Figures include the original staircase model as a special case, errors are expected to be negligible in all other Figures. The standard deviations are provided in the Source Data.

The same algorithm is used to simulate the extensions of the staircase model and the precise parameters used in simulations are detailed in the Supplementary Note 10.

### Analytical and numerical techniques
In addition to computer simulations, we developed a combination of analytical and numerical techniques, which are used to compute the evolutionary time, ecological time and the adaptation rate. In contrast to simulations, these techniques are based on closed-form solutions of master equations, numerical methods for solving algebraic equations (such as Newton-Raphson method) and solving ODEs (such as Runge-Kutta method). To illustrate these techniques, we show how to

compute the expected waiting times at a resistance state $R$ and the adaptation rate $a_R$ in the original staircase model.

1. **Find the wild-type profile $N_x$ at a resistance state $R$.** Use Newton-Raphson method to solve the algebraic equations for the non-trivial fixed point of the mean-field theory

$$0 = r(1 - N_x/K)N_x \mathbb{1}_{x \leq R} - \delta N_x \\ + \nu(N_{x-1} - N_x)\mathbb{1}_{1<x} + \nu(N_{x+1} - N_x)\mathbb{1}_{x<L}, \tag{8}$$

where $\mathbb{1}$ is the indicator function.

2. **Compute the evolutionary time $\mathbb{E}T^R_{\text{evo}}$.** A closed-form solution exists and follows from a modification of the theory in Hermsen et al.[25,26]. We consider three independent adaptation paths along which wild-type can produce a first mutant in the overlap region $x = R + 1$. On the position-genotype lattice $(x, g)$, they are the division path D: $(R, R) \to (R, R+1) \to (R+1, R+1)$ (mutation at $x = R$ followed by migration to the right); the overlap path O: $(R+1, R) \to (R+1, R+1)$ (mutation at $x = R+1$); and the no-division path N: $(R+2, R) \to (R+2, R+1) \to (R+1, R+1)$ (mutation at $x = R+2$ followed by migration to the left). As the original theory[26] assumes low motility, it neglects the no-division path since there are no wild-type cells to mutate at position $x = R + 2$ in this regime (SI Theorem 3). Moreover, we assume that the flux of mutants between neighbouring compartments in the division region $x \leq R$ and the no-divison region $x \geq R + 2$ is equilibrated, so that we can ignore the net probability flux of mutants that enter or exit the compartments $x = R - 1, R, R + 1$ and also ignore longer paths such as $(R-1, R) \to (R-1, R+1) \to (R, R+1) \to (R+1, R+1)$ (mutation at $x = R - 1$ followed by double migration to the right). This is a reasonable assumption as the flux of mutant-producing wild-type is equilibrated far from the population front. The computation of $\mathbb{E}T^R_{\text{evo}}$ has the following structure.

   (a) The waiting times till a mutant is produced in the overlap region along path $i \in \{D, O, N\}$ are denoted by $T_i$ (Fig. 2a).
   (b) The probability that the waiting will be longer than $t$ is $S_i(t) = \mathbb{P}(T_i > t)$ and the probability density function of $T_i$ is given by $F_i(t) = -dS_i/dt$.
   (c) Notice that $T^R_{\text{evo}} = \min_i T_i$ and that $T_i$ are independent. Thus, $S(t) = \mathbb{P}(T^R_{\text{evo}} > t) = S_D(t)S_O(t)S_N(t)$.
   (d) The probability density function of $T^R_{\text{evo}}$ is

$$F(t) = -\frac{dS}{dt} = F_D(t)S_O(t)S_N(t) \\ + S_D(t)F_O(t)S_N(t) \\ + S_D(t)S_O(t)F_N(t). \tag{9}$$

   (a) The expected time $\mathbb{E}T^R_{\text{evo}} = \int_0^\infty tF(t)dt$ is found by numerical integration.

   Therefore, the problem is solved by finding the functions $S_i(t)$ and differentiating them to produce $F_i(t)$. According to Hermsen et al.[25,26], these functions are:

   (a) **For path D:**

$$S_D(t) = \left(\frac{c_D e^{b_D t}}{c_D \cosh(c_D t) + b_D \sinh(c_D t)}\right)^{a_D} \tag{10}$$

where

$$a_D = \frac{\mu_f N_R}{r' - \mu_f}, b_D = \frac{1}{2}(\nu + \delta - r' + \mu_f + \mu_b), \\ c_D = \frac{1}{2}\left[\left(\nu + r' - \mu_f + \delta + \mu_b\right)^2 - 4(r' - \mu_f)(\delta + \mu_b)\right]^{1/2}, \tag{11}$$

and

$$F_D(t) = \nu S_D(t)\langle n(t) \rangle_D, \tag{12}$$

where $\langle n(t) \rangle_D$ is the average mutant population in compartment $x = R$ at time $t$

$$\langle n(t) \rangle_D = \frac{\mu_f N_R \tanh(c_D t)}{c_D + b_D \tanh(c_D t)}. \tag{13}$$

   (b) **For path O:**

$$S_O(t) = e^{-\mu_f N_{R+1} t}, \tag{14}$$

$$F_O(t) = \mu_f N_{R+1} e^{-\mu_f N_{R+1} t}. \tag{15}$$

   (c) **For path N:**

$$S_N(t) = \left(\frac{c_N e^{b_N t}}{c_N \cosh(c_N t) + b_N \sinh(c_N t)}\right)^{a_N} \tag{16}$$

where

$$a_N = -N_{R+2}, b_N = \frac{1}{2}(\nu + \delta + \mu_f + \mu_b), \\ c_N = \frac{1}{2}\left[(\nu - \mu_f + \delta + \mu_b)^2 + 4\mu_f(\delta + \mu_b)\right]^{1/2}, \tag{17}$$

and

$$F_N(t) = \nu S_N(t)\langle n(t) \rangle_N, \tag{18}$$

where $\langle n(t) \rangle_N$ is the average mutant population in compartment $x = R + 2$ at time $t$

$$\langle n(t) \rangle_N = \frac{\mu_f N_{R+2} \tanh(c_N t)}{c_N + b_N \tanh(c_N t)}. \tag{19}$$

Finally, if there is a nonnegligible chance that the first mutant in the overlap region dies before division, the evolutionary time $T^R_{\text{evo}}$ must be corrected to $T^R_{\text{evo}}/q$, where $q$ is the probability that the first mutant in the overlap region divides before its death (Supplementary Note 8).

3. **Compute the ecological time $\mathbb{E}T^R_{\text{eco}}$.** The competition between wild-type and mutants can be described well by the mean-field equations:

$$\frac{dw_x}{dt} = -\mu_f w_x + \mu_b m_x \\ + r\left(1 - \frac{w_x + m_x}{K}\right)w_x \mathbb{1}_{x \leq R} - \delta w_x \\ + \nu(w_{x-1} - w_x)\mathbb{1}_{1<x} + \nu(w_{x+1} - w_x)\mathbb{1}_{x<L} \\ \frac{dm_x}{dt} = +\mu_f w_x - \mu_b m_x \\ + r\left(1 - \frac{w_x + m_x}{K}\right)m_x \mathbb{1}_{x \leq R} - \delta m_x \\ + \nu(m_{x-1} - m_x)\mathbb{1}_{1<x} + \nu(m_{x+1} - m_x)\mathbb{1}_{x<L} \tag{20}$$

with initial conditions at time $t = 0$:

$$w_x = N_x - \langle n(\mathbb{E}T^R_{\text{evo}}) \rangle_D, x = 1, \ldots, R \\ w_x = N_{R+1} - 1, x = R + 1 \\ w_x = N_x - \langle n(\mathbb{E}T^R_{\text{evo}}) \rangle_N, x = R + 2, \ldots, L \\ m_x = \langle n(\mathbb{E}T^R_{\text{evo}}) \rangle_D, x = 1, \ldots, R \\ m_x = 1, x = R + 1 \\ m_x = \langle n(\mathbb{E}T^R_{\text{evo}}) \rangle_N, x = R + 2, \ldots, L \tag{21}$$

In these equations $w_x$ and $m_x$ stand for the number of wild-type and mutant cells in spatial compartment $x$, respectively. The system is simulated from the end state of the previous stochastic process when a first mutant appears in the overlap region. This system can be simulated efficiently by a Runge-Kutta (RK4) method. The time $t = \mathbb{E}T_{\text{eco}}^R$ is reached when $w_{R+1} < m_{R+1}$ for the first time, i.e., the mutants outcompete the wild-type in the overlap region.

4. **Find the adaptation rate**. The adaptation rate is found from

$$a_R = \frac{1}{\mathbb{E}T_{\text{evo}}^R + \mathbb{E}T_{\text{eco}}^{R-1}}. \tag{22}$$

For convenience, we refer to this set of techniques as analytical techniques, even though they implement many numerical procedures. The benefit of the analytical techniques, as opposed to simulations, comes from their computational efficacy. In addition to this form of analytical techniques, we developed a purely analytical approach, which describes the formation of the wild-type profile and this approach is detailed in the Supplementary Notes.

### Reporting summary

Further information on research design is available in the Nature Portfolio Reporting Summary linked to this article.

## Data availability

The source data generated in this study are provided in the Source Data file. Source data are provided with this paper.

## Code availability

All the code[58] used in this work is available at: https://github.com/vit-pi/StaircaseModel.

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

## Acknowledgements
We thank Kacper Kornet and Deryck Thake from the IT team of the Department of Applied Mathematics and Theoretical Physics of the University of Cambridge for their help with the managing of the super-computing cluster. We thank Julian Parkhill for discussions about the evolution of antibiotic resistance and for suggesting the study of HGT. We thank Kevin Foster, Oscar Despard, George Fortune, Pierre Haas, Julian Parkhill, Robert Austin, Mike Cates, Nick Barton and Philip Maini for their comments on earlier versions of this manuscript. V.P. was supported by a CMS Summer Research Grant, MI 2022—Oxford Mathematical Institute Scholarship, and had additional support from Gonville & Caius College. N.M.O. was supported by a BBSRC Discovery Fellowship No. BB/T009098/1.

## Author contributions
V.P. and N.M.O. designed the research. V.P. performed the research. V.P. and N.M.O. analysed the data. V.P. and N.M.O. wrote the paper.

## Competing interests
The authors declare no competing interests.
