## [Peer Review File · Nature Communications]

Bacterial motility can govern the dynamics of antibiotic resistance evolutionReviewers' Comments:

Reviewer #1:

Remarks to the Author:

Review Bacterial motility as a driver of antibiotic resistance evolution by Vit Piskovsky and Nuno M. Oliveira.

This manuscript describes the effect of motility of bacterial cells in the adaptation to spatial gradients of antibiotics. It builds on empirical and theoretical literature on the spatial evolution of antibiotic resistance, and extends that with an investigation of (heterogeneous) bacterial motility. The findings are relevant and significant, because motility/ migration is relevant in general for a wide range of (infectious) (micro)organisms, and in this context specifically for the rate of adaptation to antibiotics.

The work extensively investigates effect of several relevant motility scenarios, and even extends those to match other potentially relevant biological aspects, such as HGT. The addition of these scenarios is in line with the (main) conclusion of this manuscript.

As far as I can judge the methods are sufficiently well described and the methodology is sound. I therefore only have very few minor comments on this manuscript.

Line 39: Perhaps a little subjective, but this line sounds a little overly dramatic to me. I therefore initially took the manuscript a little less serious. (feel free to keep it in, but I am providing a reader perspective here)

'Yet, despite these important realizations, there is surprisingly little evidence about how motility contributes to bacterial adaptation, namely to the evolution of antibiotic resistance, which is ranked among other major threats such as climate change and terrorism.'

Line 162: 'At high motility ($v > \delta$, Supplementary Movie 1), the wild-type is spread across space (SI Theorem 3), competing with mutants for space even in regions where the wild-type cannot grow.'

I do not fully understand this, yet this is important for much of the manuscript. Do the authors mean that at these regions the wild-type cannot grow, but can be maintained and therefore contributes to K, and therefore is able to compete for space with mutants? Can the authors explain/ discuss?

Figure 1b:

Blue and red are only explained at panel b, but are already described at panel a. Therefore suggest to move up to panel a.

Figure 1d: Nature of the red bars is unclear. Therefore suggest to add description to caption.

Line 247: Replace in 'this' case for in 'the latter' case for clarification.

Line 323: 'This result stems from the fact that the slower phenotype is naturally selected for in the absence of phenotypic switching, given that the slower phenotype is exposed to antibiotics less often and maximises the local fitness of the wild-type, and stochastic switching levels out the numbers of both phenotypes'

Clarification needed: Why would the slower phenotype be less exposed to antibiotics?

Line 368: 'As a result, the population has decreased antibiotic exposure and adaptation rate.'

Yet, if motility would be sufficiently fast, then the antibiotic exposure may be averaged at short time-scale. Correct? (I do understand that average-step sizes are not in the staircase model) but would that change dynamics by a different (likely larger) mutational supply, and (likely lower) cost of resistance? The authors refer to these aspects in the last part of the discussion, and I wonder whether time averaged concentrations would therefore change the effect of motility as described in line 368.

Reviewer #2:

Remarks to the Author:

Piskovsky and Oliveira used a stochastic mathematical model to study the interaction of cell motility and bacterial adaptation in a spatially heterogeneous environment. In particular, they use an adapted version of the staircase model of Hermsen et al (a lattice of genotypes and spaces with increasing antibiotic concentrations) and show that increasing the migration rate drastically changes the dynamics of the original model, since not only evolutionary time (defined as the earliest time a more resistance mutant is produced) but also ecological time (defined as the earliest time when a mutant population becomes larger than the wild-type in the overlap region) becomes relevant for adaptation. They further depict how other characteristics of active bacterial motility, i.e. phenotype switching between high and low motility and chemotaxis, can change adaptive rates in more complex models. They describe 3 different motility regimes and show that high motility can lead to population extinction in a sink-like environment. They conclude that motility can also limit adaptation, an effect which has been overseen in the original staircase model.

The modeling/analysis in this work is of high quality, and its predictions could be a good foundation to assess the effect of motility on resistance evolution experimentally. However the biological relevance of the model remains unclear. In addition, some of the outcomes of the analysis are somehow trivial.

Major points

1. It is unclear what the relationship between v and δ represents in the physical world and when, in practice, should we expect transitions between high and low motility scenarios. What are the relevant units (in the real world) and how can they be compared, especially in a continuous space? How to know (in a real experiment) which regime applies?

2. The simulation results need error bars, e.g. in Figure 2. It is unclear when the decrease/increase in adaptation rate is actually significant, especially when analytics predict a steep reduction.

3. When evaluating the effect of cell-density dependent switching on adaptation rates, the authors consider all cells in a single compartment to switch their phenotype. This leads to dynamics where swarming populations (in the fast to slow scenario) might not switch phenotype ever again and just spread into all the compartments at low density, which is a rather trivial outcome. Would the author's conclusions still hold true in a model where the random switching rate s is expressed as a function of cell density N , and where not all cells but only a subpopulation in a compartment undergoes this switch (similarly to Fig. 3). This would also much better reflect phenotypic heterogeneity in biological systems, where both phenotypes usually coexist in the same space.

4. Growth in each compartment is dependent on the presence of all cells in this compartment, whether they are dividing or not. In nature, not growing cells might not compete fully for resources with growing cells. The authors have taken this assumption from the original staircase model, which is adequate for showing differences with the original model, but to make the current model biologically

more meaningful it would be important to explore how differential competition between growing and non growing cells might interfere with the dynamics they have described.

5. The title should indicate that this work is exclusively numerical/theoretical.

Minor points

6. Several important parameters, such as the mutation rate (which is set to be different by 3 magnitudes between increase and decrease of resistance without justification) are identical to the original work by Hermsen et al.. While this is adequate to show differences with the original staircase model, it would be good if they could at least comment on this choice, and in the best case also look at its interaction with motility.

7. The authors should comment on the ratio of different parameters that can lead to the different motility regimes they have described, and how these ratios relate to data from biological/experimental systems, and if they are biologically meaningful.

8. Mutant fitness (calculated as $r(1-Nx/K)-\delta$) is actually expected growth dependent on cell density, and does in my opinion not really correspond to the usage of this term in experimental work. The fitness gradient therefore rather represents the steepness of the population distribution. Another term or a comment in the text would be needed to make this less misleading.

9. The graphical representations of the model, especially the depiction of the heat maps (usually in Fig. Xb on the left) would benefit from more clarity. x and y scales should be uniform within a figure. The comments on Figure 5 ("increase time" and "increase motility") in combination with arrows are misleading for the understanding of the Figure.

Reviewer #3:

Remarks to the Author:

Review: Piskovsky & Oliveira

This manuscript reports results on (versions of) a mathematical / computational model. It aims to study the effects of several aspects of motility on the rate of antibiotic resistance evolution in an environment that contains a concentration gradient of an antibiotic.

The model used is based on the so-called Staircase Model published in 2012 in PNAS (Hermsen *et al*), which assumes a one-dimensional environment that is subdivided into compartments with increasing drug concentrations. In the article, the analysis of the model was restricted to a particular parameter regime: it assumed that the carrying capacity was large, the migration rate between neighboring compartments low, and mutation rates lower still. The current manuscript expands this analysis by studying the regime of fast migration/motility. Next, the authors expand the model by including stochastic switching between a fast and a slow motility state, or a density-dependent motility rate.

As I see it, the main findings are presented in the first four figures. Figure 1 demonstrates that the behavior of the high-motility regime is qualitatively different from that of the low-motility regime: it results in smoother density profiles, which reduces the growth rate of mutants with a higher level of resistant emerging in the system and hence increases the “ecological waiting time”. Figure 2 shows that, as a consequence, the adaptation rate depends non-monotonously on the motility rate, with a positive dependence in the low motility regime but a negative dependence in the high-motility regime. Figure 3 presents results of the model in which individual cells switch stochastically between two motility rates. The main point is that this extended model behaves very similarly to the original model with some effective migration rate. The same holds for the model with density-dependent motility, as presented in Figure 4.

Apart from these main points presented in full in the main text, quite a lot of additional work has been shifted to supplementary figures, where the effect of a fitness cost, chemotaxis, HGT, different types of antibiotics, etc. are also examined, and to the supplementary text, which contains detailed mathematical derivations and analysis.

Generally, I think the analysis is sound and the conclusions seem warranted. The text is clearly and attractively written and the figures are carefully designed. I only have minor comments.

Comments on content

1. Around the same time when the Staircase Model was published, Greulich et al (PRL, 2012) published a similar model. An important difference between Hermsen et al and Greulich et al was that Hermsen et al focused on a series of relatively isolated compartments, whereas Greulich et al considered continuous space. It seems to me that this difference disappears in the high-motility regime. While the authors cite Greulich et al, they do not compare their results to that paper in much detail. I think they should.
2. Section D and Figure 5 of the Results section discusses the effect of migration on the density profile from a dynamic systems perspective. Although the mathematics presented here and in the SI are fun (for some readers), the ultimate biological messages are very intuitive and can easily be summarized in a few sentences, as the authors in fact do in the last paragraph of this section. I would advice to move this part to the supplement.
3. How did the authors calculate the adaptation rates of, say, Fig. 2b in their simulations? If I understand well, they used the equation $a_R = 1/(\mathbb{E}\{T_{\text{evo}}\} + \mathbb{E}\{T_{\text{eco}}\})$ and measured T_{evo} and T_{eco} in their simulations. I would argue it makes more sense to measure the actual time it took in the simulations to take a certain number of steps. The authors argue that this should equal the equation above, but this is only approximately so. In the current setup, this assumption remains untested.
4. In introduction and discussion you claim that “cell motility has largely been overlooked”. I think this is claim is a bit dubious given that all mathematical models that you cite do include motility, albeit at a fixed and relatively low rate. Perhaps you can make this claim more specific.

Comments on figures

1. In several figures, genotype is indicated with a blue or yellow color scale. Even though genotype is restricted to integers, the legend belonging to the color scale shows integers at the *border* between the colors. This is confusing.
2. In the three panels of Fig. 3b, the lines indicating the borders between the regimes seem to be inconsistent. In particular, in the heat maps the horizontal and vertical black lines are not at $\nu_i = \delta = 10^{-1}$.
3. I would prefer that the parameters used in the figures are mentioned in the figure caption.

Comments on text and notation

1. What is the rationale behind the term "overlap region"? Overlap between what?
2. In several places (both in the main text and in the supplement), assumptions are made that are not explicitly mentioned. As mentioned above, the equation $a_R = 1/(\mathbb{E}T_{\text{evo}}) + \mathbb{E}T_{\text{eco}}$ is only approximately true. Also, I don't think it is mentioned anywhere that the analytical results rely on the assumption that K is large (at least, I think they do) and that the mutation rates are low. (Without these assumptions, I do not expect the mean-field equations and quasi-steady-state assumptions to work.) Please check the text for implicit assumption.
3. The notation of the expected value in the equation $a_R = 1/(\mathbb{E}T_{\text{evo}}) + \mathbb{E}T_{\text{eco}}$ in the main text is not introduced.
4. On line 195, the authors write "low motility ($\nu < \delta$) accelerates the adaptation rate". Similar phrases appear later in the manuscript, including on line 216 ("high motility (...) decelerates the adaptation rate"). These phrases are confusing because in fact the adaptation rate is much lower in the low-motility regime than in the high-motility regime. I would rephrase this.
5. line 290: "to to"
6. line 328: What is meant by "stochastic switching levels out the numbers of both phenotypes"?
7. line 479: "Similar optimum" -> "A similar optimum"

Comments from reviewers

REVIEWER 1

1804
1805
1806 Review Bacterial motility as a driver of antibiotic resistance evolution by Vit Piskovsky and Nuno M. Oliveira.
1807 This manuscript describes the effect of motility of bacterial cells in the adaptation to spatial gradients of antibiotics.
1808 It builds on empirical and theoretical literature on the spatial evolution of antibiotic resistance, and extends that with
1809 an investigation of (heterogeneous) bacterial motility. The findings are relevant and significant, because motility/
1810 migration is relevant in general for a wide range of (infectious) (micro)organisms, and in this context specifically for
1811 the rate of adaptation to antibiotics.
1812 The work extensively investigates effect of several relevant motility scenarios, and even extends those to match
1813 other potentially relevant biological aspects, such as HGT. The addition of these scenarios is in line with the (main)
1814 conclusion of this manuscript.
1815 As far as I can judge the methods are sufficiently well described and the methodology is sound. I therefore only
1816 have very few minor comments on this manuscript.

A1. Comments

1817
1818 Line 39: Perhaps a little subjective, but this line sounds a little overly dramatic to me. I therefore initially took the
1819 manuscript a little less serious. (feel free to keep it in, but I am providing a reader perspective here) ‘Yet, despite these
1820 important realizations, there is surprisingly little evidence about how motility contributes to bacterial adaptation,
1821 namely to the evolution of antibiotic resistance, which is ranked among other major threats such as climate change
1822 and terrorism.’

1823 This paragraph is now rephrased in the revised manuscript to “Despite these important realizations, we know little
1824 about how cell motility contributes to bacterial adaptation, namely to the evolution of antibiotic resistance, which is
1825 a major public health concern [8].”
1826

1827
1828 Line 162: ‘At high motility ($\nu > \delta$, Supplementary Movie 1), the wild-type is spread across space (SI Theorem 3),
1829 competing with mutants for space even in regions where the wild-type cannot grow.’ I do not fully understand this,
1830 yet this is important for much of the manuscript. Do the authors mean that at these regions the wild-type cannot
1831 grow, but can be maintained and therefore contributes to K , and therefore is able to compete for space with mutants?
1832 Can the authors explain/discuss?

1833 In our model, we assume logistic growth in each compartment. Importantly, even in regions where wild-type
1834 cells cannot grow, they can be continually imported from neighbouring regions and, consequently, occupy space and
1835 contribute towards the carrying capacity K . As a result, the presence of wild-type cells decreases the growth of
1836 mutant cells. We have modified line 162 so that this idea is now clearer. As another reviewer also asked about this
1837 issue, and in particular about the plausibility of assuming equal competitive ability between growing and non-growing
1838 cells, we have now considered the effect of non-growing cells that compete at a reduced rate with growing cells (new
1839 Supplementary Text S4, new Supplementary Fig. 4, shown here as Fig. A1). Fig. A1 shows that differential
1840 competition between growing and non-growing cells does not affect the key conclusions of our work. We now mention
1841 this effect in Results section B, and in the Discussion.
1842

1843
1844 Figure 1b: Blue and red are only explained at panel b, but are already described at panel a. Therefore suggest to
1845 move up to panel a.

1846 We had a typo in the caption of the figure, namely (black dots) instead of (blue and red dots). To clarify, dots in
1847 panel a represent cells, and we do not explain the meaning of specific colours. The meaning of the colours (i.e., why
1848 we distinguish between blue and red dots) only becomes important when we explain the adaptation process. For this
1849 reason, we postpone the explanation of the colour scheme to panel b of the figure.
1850

1851
1852 Figure 1d: Nature of the red bars is unclear. Therefore suggest to add description to caption.
1853 Added.
1854
1855

Figure A1 | Differential competition between growing and non-growing cells. In our original model, growing and non-growing cells are assumed to contribute equally to the carrying capacity. Here, we consider the effect of non-growing cells contributing to the carrying capacity at a reduced rate $\alpha \in [0, 1]$ when compared to growing cells ($\alpha = 0$ corresponds to no contribution, $\alpha = 1$ corresponds to the same contribution as in the original model). Therefore, the division rate of growing cells ($g \geq x$) is $\max(0, r(1 - (N_x^g + \alpha N_x^n)/K))$, where N_x^g is the number of growing cells above the staircase ($g \geq x$) and N_x^n is the number of non-growing cells below the staircase ($g < x$) in a given spatial compartment x . The heatmap of adaptation rate on the (ν, α) plane explains how the reduction in competitive strength of non-growing cells α changes the relationship between motility rate ν and adaptation rate a_R . The data shows that there is no significant change in the relationship between adaptation rate and motility. A detailed description of this model can be found in Supplementary Text S4.

1856 Line 247: Replace in ‘this’ case for in ‘the latter’ case for clarification.

1857 Replaced.

1858

1859

1860 Line 323: ‘This result stems from the fact that the slower phenotype is naturally selected for in the absence of
1861 phenotypic switching, given that the slower phenotype is exposed to antibiotics less often and maximises the local
1862 fitness of the wild-type, and stochastic switching levels out the numbers of both phenotypes’ Clarification needed:
1863 Why would the slower phenotype be less exposed to antibiotics?

1864 The slower phenotype is less exposed to antibiotics because slower cells are less likely to migrate into regions with
1865 large antibiotic concentrations, and their population will be concentrated in regions where they can divide. This
1866 argument is consistent with the theory developed by Alan Hastings in [A1], where the author shows that spatial
1867 heterogeneity alone (such as the one introduced by a stable antibiotic gradient) selects against the evolution of
1868 dispersal/migration/motility. In particular, the author argues that slower phenotypes are better at staying in regions
1869 with high fitness, such as low antibiotic concentration in our case. Moreover, we note that our equation for the wild-
1870 type profile (S16) is of the same form of equation (1) in Hastings’s paper if the switching rate s is 0. Also, our Fig.
1871 1d shows that, at lower motility, the population is concentrated in compartments where it can divide. In this sense,
1872 the slower phenotype is less exposed to antibiotics. We modified the original line 323 to make this idea clearer: “This
1873 result stems from the fact that the slower phenotype is naturally selected for in the absence of phenotypic switching,
1874 given that the slower phenotype spends more time in regions of low antibiotic concentration where it can proliferate,
1875 ...”.

1876

1877

1878 Line 368: ‘As a result, the population has decreased antibiotic exposure and adaptation rate.’ Yet, if motility would
1879 be sufficiently fast, then the antibiotic exposure may be averaged at short time-scale. Correct? (I do understand that
1880 average-step sizes are not in the staircase model) but would that change dynamics by a different (likely larger)
1881 mutational supply, and (likely lower) cost of resistance? The authors refer to these aspects in the last part of
1882 the discussion, and I wonder whether time averaged concentrations would therefore change the effect of motility as
1883 described in line 368.

1884 To clarify, line 368 was used in the context of density-dependent motility, with cells moving fast at low density
1885 (rate ν_L) and slowly at high density (rate ν_H). In this case, the vast majority of the population reaches high density,

1886 moves slowly and resides in low antibiotic concentrations (Fig. 4d, lower right panel). Therefore, the time-averaged
 1887 antibiotic concentrations of the entire population are approximated by the time-averaged antibiotic concentrations
 1888 of the high-density phenotype, which are low and independent of the time-averaged antibiotic concentrations of the
 1889 few fast cells. This pattern is reflected in the idea of effective motility, where the system behaves as if only the
 1890 high-density phenotype was present (with motility ν_H). It should be noted also that line 368 is used in a very specific
 1891 context, namely to highlight exceptions to the rule of effective motility: “First, when cells switch from fast to slow
 1892 above the threshold S , the population becomes very stable as the fast cells at the front quickly return to the slow
 1893 population bulk (Fig. 4c). As a result, the population has decreased antibiotic exposure and adaptation rate (Fig.
 1894 4b).” Therefore, line 368 simply meant to highlight that the population adapts slower than a population with single
 1895 motility ν_H in the original staircase model. Regarding the effects of resistance costs and mutational supply, if we
 1896 understood correctly, the reviewer asks if an increased mutation rate μ_f and decreased cost of resistance c can change
 1897 this comparison between the original model and the model with density-dependent motility. To clarify this issue, we
 1898 would like to note that in the density-dependent motility model, we assumed $c = 0$ and thus resistance costs cannot
 1899 be further reduced. Moreover, the mutation rate does not affect the reduction in cell number at the population front
 1900 of the wild-type profile, which can be concluded from the absence of μ_f parameter in the wild-type profile equations
 1901 (S6) and (S29). Therefore, we believe that reducing c and or increasing μ_f do not change the idea expressed in our
 1902 original line 368.

1903 We would like to thank the reviewer for this question because it made us realize one aspect that we did not consider
 1904 originally. While we studied the effects of resistance costs (Supplementary Fig. 2), this question made us wonder
 1905 about the effects of mutation rates μ_f . We have now varied the mutation rate μ_f (new Supplementary Fig. 3, Fig.
 1906 A2 here) and showed that our key conclusions are not affected by it for realistic values of mutation rate, and that the
 1907 low, high and deadly motility regimes capture well the evolutionary dynamics.

Figure A2 | Effect of mutation rate. The forward mutation rate μ_f (towards increasing resistance) is varied in the original model. The heatmap of adaptation rate on the (ν, μ_f) plane explains how the mutation rate μ_f changes the relationship between motility rate ν and adaptation rate a_R . Mutation rate below $\mu_f < 10^{-4}$ does not affect the relationship between adaptation rate and motility: adaptation rate increases at low motility, decreases at high motility, and no adaptation occurs at deadly motility (grey). This mutation rate threshold corresponds to the probability of $\mu_f/\delta = 10^{-3}$ resistance mutations per cell division, which is very high compared to experimental estimates of 10^{-6} and 10^{-9} mutations per cell division [A2–A4].

1908 **REVIEWER 2 (REMARKS TO THE AUTHOR):**

1909 Piskovsky and Oliveira used a stochastic mathematical model to study the interaction of cell motility and bacterial
 1910 adaptation in a spatially heterogeneous environment. In particular, they use an adapted version of the staircase
 1911 model of Hermsen *et al* (a lattice of genotypes and spaces with increasing antibiotic concentrations) and show that
 1912 increasing the migration rate drastically changes the dynamics of the original model, since not only evolutionary time
 1913 (defined as the earliest time a more resistance mutant is produced) but also ecological time (defined as the earliest time
 1914 when a mutant population becomes larger than the wild-type in the overlap region) becomes relevant for adaptation.
 1915 They further depict how other characteristics of active bacterial motility, i.e. phenotype switching between high and
 1916 low motility and chemotaxis, can change adaptive rates in more complex models. They describe 3 different motility
 1917 regimes and show that high motility can lead to population extinction in a sink-like environment. They conclude that
 1918 motility can also limit adaptation, an effect which has been overseen in the original staircase model.

1919 The modeling/analysis in this work is of high quality, and its predictions could be a good foundation to assess
 1920 the effect of motility on resistance evolution experimentally. However the biological relevance of the model remains
 1921 unclear. In addition, some of the outcomes of the analysis are somehow trivial.

1922 **A2. Major points**

1923 1. It is unclear what the relationship between ν and δ represents in the physical world and when, in practice,
 1924 should we expect transitions between high and low motility scenarios. What are the relevant units (in the real world)
 1925 and how can they be compared, especially in a continuous space? How to know (in a real experiment) which regime
 1926 applies?

1927 Our main purpose was to develop theory on how cell motility affects bacterial adaptation in spatially-heterogeneous
 1928 environments and we use the evolution of antibiotic resistance as a case study. While our goal was to provide a general
 1929 theory that was not meant to recapitulate any experimental setup in particular, we understand the concern of the
 1930 reviewer and the importance of relating our model results to experimental studies. We now created a new supplement-
 1931 ary section in the paper that addresses this issue (new Supplementary Text S7). In particular, we now considered
 1932 what we call a visiting number, a non-dimensional number that captures the antibiotic variability experienced by an
 1933 average cell during its lifetime. Specifically, the visiting number V is defined as the number of regions that differ in
 1934 antibiotic concentrations (on MIC scale, MIC = minimal inhibitory concentration) and that are visited by an average
 1935 cell during its lifetime. This number can be computed for both our model and different experimental setups, and
 1936 we now examine the visiting number in our model and in two works, [A5] and [A6], that studied experimentally the
 1937 evolution of antibiotic resistance in heterogeneous environments. In our model, the visiting number is $V = \nu/\delta$ as ν
 1938 is the number of compartments visited per unit of time and $1/\delta$ is the timescale of bacterial lifespan. In experiments,

$$V = \frac{vt}{l},$$

1939 where v is the characteristic cell speed, t is the doubling time and l is the length-scale over which drug concentrations
 1940 vary on MIC scales. This expression follows as v/l estimates the number of different MIC scales visited per unit
 1941 of time and t characterises the timescale of bacterial lifespan. The visiting number can be used to identify the
 1942 adaptation regime since Fig. 2b shows that the low motility regime corresponds to $\nu < \delta$ (i.e., $V < 1$) and the high
 1943 motility regime to $\nu > \delta$ (i.e., $V > 1$). To relate our model to experiments, we computed the visiting numbers in the
 1944 experiments [A5, A6] and noted the predicted motility regime in Table I (new Supplementary Table I). To further
 1945 confirm such prediction, we compared the qualitative features of the experimental evolutionary dynamics with the
 1946 results of our model in Fig. 1d. In [A5] ($V = 120$), the resident mutants quickly invaded the entire environment,
 1947 which is characteristic of the high motility regime in Fig. 1d. In [A6] ($V = 0.1846$), the adaptation dynamics was
 1948 located at the population front and multiple strains with different resistance levels coexist, which is characteristic of
 1949 the low motility regime in Fig. 1d. This comparison suggests that the key results of our model can be compared with
 1950 experimental studies. Notably, our new analysis predicts that bacteria in [A5] and [A6] were evolving in very different
 1951 motility regimes, which may explain why the authors observed different adaptation rates in their studies. Find more
 1952 details in Discussion and in Supplementary Text S7.

1953

1954

1955 2. The simulation results need error bars, e.g. in Figure 2. It is unclear when the decrease/increase in adaptation
 1956 rate is actually significant, especially when analytics predict a steep reduction.

1957 We now added error bars to Fig. 2b and showed that they are smaller than the symbols in Fig. 2b, see Fig. A3,

TABLE I | Experimental examples of different adaptation regimes explored in our work.

experiment	v	t	l	V	adaptation regime	qualitative evidence for the regime
MEGA-plate [A6]	10 μ m/s [A7, A8]	40min [A9, A10]	13cm	0.1846	low motility regime	adaptation dynamics located at the front, coexistence of resistance strains
microenvironments [A5]	10 μ m/s [A7, A8]	40min [A9, A10]	200 μ m	120	high motility regime	resistant mutants invade the entire environment and replace the wild-type

1958 consistent with what was found in [A11]. We now clarified this observation in the caption of Fig. 2. To clarify, the
 1959 central limit theorem guarantees a small size of the errors for a sufficiently large number of simulations n (see a detailed
 1960 description of computation below). In Fig. 2b, we use $n = 50$ simulations and we show that this number is sufficient
 1961 to guarantee that error bars are small for all motility values. As we use $n = 50$ simulations in all Figures and all
 1962 Figures include the original staircase model as a special case, errors are expected to be negligible in all other Figures.
 1963 We now explicitly list all errors in the new Source Data file and clarify this effect in Methods. For completeness, we
 1964 add below the description of the computation.

1965 *Detailed computation of the error bars.* Let us denote the observed times between two adaptation steps in a
 1966 simulation i by $\Delta T_i^R = T_i^R - T_i^{R-1}$. The equation for estimating the adaptation rate (5) is

$$a_R = \frac{n}{\sum_i \Delta T_i^R}.$$

1967 Defining a_R in this way, we can notice that the central limit theorem implies that

$$\sqrt{n}(a_R^{-1} - \mathbb{E}\Delta T_1^R) \rightarrow N(0, \text{Var}(\Delta T_1^R)).$$

1968 Using the delta method for $g(x) = x^{-1}$, we can see that

$$\sqrt{n}(a_R - (\mathbb{E}\Delta T_1^R)^{-1}) \rightarrow N(0, \text{Var}(\Delta T_1^R)/(\mathbb{E}\Delta T_1^R)^4).$$

1969 Therefore, we can use the sample moments

$$\mu = \frac{\sum_i \Delta T_i^R}{n}$$

$$\sigma^2 = \frac{\sum_i (\Delta T_i^R - \mu)^2}{n}$$

1970 to compute the standard deviation associated with estimating the adaptation rate a_R by finite number n of simulations

1971 as

$$\text{sd}(a_R) = \sqrt{\frac{\sigma^2}{n\mu^4}}.$$

1972 Using this formula, we plotted the standard deviation as error bars centred at the symbols that denote the estimated
 1973 mean value of the adaptation rate a_R (Fig. A3). We can see that the error bars are so small that they are only visible
 1974 after sufficient zooming into the figure, which is ensured by taking sufficiently large $n = 50$.

Figure A3 | Standard deviations of the estimated adaptation rate a_R . The standard deviations associated with estimating the adaptation rate a_R by finite number $n = 50$ of simulations can be computed using the central limit theorem. We plotted the standard deviation as error bars centred at the symbols that denote the estimated mean value of the adaptation rate a_R and that are used in Fig. 2b.

1975 3. When evaluating the effect of cell-density dependent switching on adaptation rates, the authors consider all cells
 1976 in a single compartment to switch their phenotype. This leads to dynamics where swarming populations (in the fast to
 1977 slow scenario) might not switch phenotype ever again and just spread into all the compartments at low density, which
 1978 is a rather trivial outcome. Would the author’s conclusions still hold true in a model where the random switching
 1979 rate s is expressed as a function of cell density N , and where not all cells but only a subpopulation in a compartment
 1980 undergoes this switch (similarly to Fig. 3). This would also much better reflect phenotypic heterogeneity in biological
 1981 systems, where both phenotypes usually coexist in the same space.

1982 For simplicity, we modelled phenotypes implicitly (i.e., all cells in a given compartment have the same phenotype).
 1983 One of the advantages of this modelling is that it reduces computational time. However, to address the comment of
 1984 the reviewer, we now show that our model can be derived as a limit of a model suggested by the reviewer, where
 1985 both phenotypes are explicitly tracked in each compartment (Fig. A4a). In particular, we show that the evolutionary
 1986 dynamics is governed by the same effective motility in both models and our key conclusions hold as originally presented
 1987 (Fig. A4b-f). The alternative modelling technique is now described in Results section C, new Supplementary Fig. 8
 1988 and new Supplementary Text S6. For completeness, we give below a detailed account of the correspondence between
 1989 the two models of density-dependent switching (Supplementary Text S6).

1990 *Details on the correspondence between the models.* First, we construct the explicit model suggested by the reviewer.
 1991 In the explicit model, we consider the same setup as in the model for stochastic switching (Fig. 3a) where two
 1992 phenotypes of different motility $\nu_{1,2}$ are considered and switch at rate s . To introduce density-dependent motility, we
 1993 modify the switching rates s between the motility phenotypes P (Fig. A4a). In particular, at a given spatial position
 1994 x with N_x cells, phenotype 1 switches into phenotype 2 at a rate

$$s_{1 \rightarrow 2} = \begin{cases} s\beta/(1 + \beta) & \text{if } N_x < S, \\ s/(1 + \beta) & \text{if } N_x \geq S, \end{cases} \quad (\text{A1})$$

1995 and phenotype 2 switches into phenotype 1 at a rate

$$s_{2 \rightarrow 1} = \begin{cases} s/(1 + \beta) & \text{if } N_x < S, \\ s\beta/(1 + \beta) & \text{if } N_x \geq S. \end{cases} \quad (\text{A2})$$

1996 The total switching rate $s_{1 \rightarrow 2} + s_{2 \rightarrow 1} = 2s$ is kept constant and the phenotypes switch β -times more likely in the
 1997 direction $1 \rightarrow 2$ than $2 \rightarrow 1$ at low density ($N_x < S$), while they switch β -times more likely in the direction $2 \rightarrow 1$ than
 1998 $1 \rightarrow 2$ at high density ($N_x > S$). To identify phenotype 1 with the low-density phenotype ($\nu_1 = \nu_L$) and phenotype
 1999 2 with the high-density phenotype ($\nu_2 = \nu_H$), we restrict to $\beta \leq 1$. Moreover, for switching to be important, we
 2000 assume that $s \gg \delta$. We can notice that the implicit model of density-dependent switching corresponds to $s \rightarrow \infty$ and
 2001 $\beta \rightarrow 0$. Therefore, the implicit model is expected to exhibit the same dynamics as the explicit model. We show this
 2002 correspondence in Fig. A4b-f. The adaptation regime is again governed by the same effective motility $\nu_{\pm} = \nu_{H/L}$ (Fig.
 2003 A4b-e) and the wild-type profiles are very similar and differ only in predicted phenotypic proportions at individual
 2004 spatial positions between the explicit and implicit models (Fig. A4f). Both models give rise to the same mutant
 2005 fitness (red bars), which further explains the similarity of the evolutionary dynamics in the two alternative models.

Figure A4 | Further effects of density-dependent motility. (a) Explicit model. In the main text, we model motility phenotypes implicitly as all cells at a given spatial position have the same motility (implicit model). Alternatively, motility phenotypes can be modeled explicitly at each spatial position x (explicit model). In this case, motility phenotypes are represented by a new dimension in the staircase model and switch at a large rate ($s \gg \delta$) with a bias towards low-density (resp. high-density) phenotype at low (resp. high) density (characterised by $\beta \ll 1$), where low/high density is determined by comparing the density of cells N_x at a spatial position x to the switching threshold S . See Supplementary Text S6 for details. (c,e) Adaptation in source-like environment. Adaptation rate heatmap on the (ν_L, ν_H) plane for different switching thresholds S in the source-like environment, where $\nu_{L,H}$ are the low-density/high-density motility rates. The (ν_L, ν_H) plane can be partitioned into different combinations of adaptation regimes and its diagonal corresponds to a population of a single motility. The diagonal separates slow-to-fast and fast-to-slow switching combinations, which differ in relative motility at low-to-high density. Bacteria generically adapt as if all cells had the same effective motility, which corresponds to the intersection of the level sets (white dashed lines) with the diagonal. This effective motility generically matches the low-density (resp. high-density) motility at high (resp. low) threshold S . At low threshold S , this generic rule has one exception in the implicit model. The adaptation rate is reduced in the mixed motility combination of the fast-to-slow switching case, as all cells in the overlap region switch to fast phenotype and are unlikely to start a growing mutant colony there. (b,d) Adaptation in sink-like environment. Same as panels (c,e), but for a sink-like environment. Importantly, the same effective motility governs the adaptation dynamics as in panels (c,e). At low threshold S , this generic rule of effective motility has three exceptions. First, the adaptation rate is reduced in the mixed motility combination of fast-to-slow switching. Second, the adaptation rate is reduced for slow-to-fast switching when the environment is sink-like and the high-density motility phenotype moves above the critical motility. Third, a deadly motility regime (grey) can only occur if the initial population is of low density and the low-density motility phenotype moves above the critical motility. In the explicit model, the critical motility is slightly increased as the explicit presence of a few slow cells with a high-density phenotype can prevent extinction. (f) Wild-type profiles in the implicit and explicit model. The wild-type profiles are similar and differ only in predicted phenotypic proportions. Note that the mutant fitness (red bars) is similar in both models, which further explains why both models exhibit the same adaptation dynamics.

2006 4. Growth in each compartment is dependent on the presence of all cells in this compartment, whether they are
 2007 dividing or not. In nature, not growing cells might not compete fully for resources with growing cells. The authors
 2008 have taken this assumption from the original staircase model, which is adequate for showing differences with the
 2009 original model, but to make the current model biologically more meaningful it would be important to explore how
 2010 differential competition between growing and non-growing cells might interfere with the dynamics they have described.

2011 We already answered a similar question from reviewer 1 and we now consider growing and non-growing cells with
 2012 different competitive ability. In short, our model assumes logistic growth and thus, even in regions where wild-type
 2013 cells cannot grow, they can be continually imported from neighbouring regions, and consequently occupy space, which
 2014 contributes towards the carrying capacity K . To understand the effect of differential competition between growing
 2015 and non-growing cells suggested by the reviewer, we now considered a model where only a proportion α of the non-
 2016 growing cells contributes to the carrying capacity, see Fig. A1 (Supplementary Text S4, Supplementary Fig. 4). Fig.
 2017 A1 shows that differential competition between growing and non-growing cells does not affect our key conclusions.

2018

2019

2020 5. The title should indicate that this work is exclusively numerical/theoretical.

2021 Our intention was to offer a title that can be understood as a new hypothesis, and not to highlight the specific
 2022 approach we used to test the hypothesis. We followed closely the titles of important works in the field, for example
 2023 “On the rapidity of antibiotic resistance evolution facilitated by a concentration gradient,” “Mutational Pathway
 2024 Determines Whether Drug Gradients Accelerate Evolution of Drug-Resistant Cells,” “Acceleration of emergence of
 2025 bacterial antibiotic resistance in connected microenvironments” or “Spatiotemporal microbial evolution on antibiotic
 2026 landscapes.” Some of these works are mathematical and others are experimental, and they do not distinguish the
 2027 specific methodology used. Therefore, we are keen to keep the title as it is.

2028

A3. Minor points

2029 6. Several important parameters, such as the mutation rate (which is set to be different by 3 magnitudes between
 2030 increase and decrease of resistance without justification) are identical to the original work by Hermsen *et al.* While
 2031 this is adequate to show differences with the original staircase model, it would be good if they could at least comment
 2032 on this choice, and in the best case also look at its interaction with motility.

2033 We have assumed the mutation rate used by Hermsen *et al.*, indeed to make our results on the role of bacterial
 2034 motility comparable to theirs, and because the choice of mutation rate was consistent with experimental evidence
 2035 [A2–A4]. However, we do appreciate the concern of the reviewer and we now added a new figure where we show
 2036 the effect of varying the mutation rate on the evolutionary dynamics of our model. See Supplementary Fig. 3. The
 2037 reviewer can also see this figure in our answer to a question posed by reviewer 1, Fig. A2. Essentially, we show that
 2038 for realistic mutation rates, our conclusions hold.

2039

2040

2041 7. The authors should comment on the ratio of different parameters that can lead to the different motility regimes
 2042 they have described, and how these ratios relate to data from biological/experimental systems, and if they are
 2043 biologically meaningful.

2044 We show in Fig. 2 that the low motility regime appears for $\nu < \delta$ and the high motility regimes appears for $\nu > \delta$.
 2045 Therefore, the ratio ν/δ describes the motility regime in our model. Moreover, in our answer to Major Points question
 2046 1 of the reviewer, we show that this ratio can be interpreted as a visiting number V and we relate this number to
 2047 experimental studies [A5, A6]. This new metric then suggests that the motility regimes described in our work are
 2048 biologically meaningful.

2049

2050

2051 8. Mutant fitness (calculated as $r(1 - N_x/K) - \delta$) is actually expected growth dependent on cell density, and does in
 2052 my opinion not really correspond to the usage of this term in experimental work. The fitness gradient therefore rather
 2053 represents the steepness of the population distribution. Another term or a comment in the text would be needed to
 2054 make this less misleading.

2055 In our work, mutant fitness follows the classic Malthusian fitness, which is frequently used to experimentally
 2056 determine bacterial fitness [A12]. To clarify, Malthusian fitness is defined as the net growth rate during the exponential
 2057 phase of a population. In population genetics, for example, the alternative definition of absolute fitness (the expected
 2058 number of adult offspring per generation) is often used [A13]. As Malthusian fitness is often considered a better metric
 2059 for models with continuous time [A14], and our model is formulated in continuous time, we use the definition of the
 2060 Malthusian fitness. We added a comment to clarify this definition to Results section A, where we define mutant fitness.

2061 We should note that there is no density-dependence in our definition of fitness in the usual sense of dependence on the
2062 density of the phenotype whose fitness is measured. This is because the term N_x in the mutant fitness $r(1 - N_x/K) - \delta$
2063 corresponds to the total number of wild-type cells rather than mutant cells. The wild-type cells contribute to the
2064 carrying capacity K and modify the environment experienced by mutant cells, and therefore, modify the net growth
2065 of mutant cells during their exponential growth phase.

2066

2067

2068 9. The graphical representations of the model, especially the depiction of the heat maps (usually in Fig. Xb on the
2069 left) would benefit from more clarity. x and y scales should be uniform within a figure. The comments on Figure
2070 5 (“increase time” and “increase motility”) in combination with arrows are misleading for the understanding of the
2071 Figure.

2072 We now changed the colour maps to avoid confusion between phenotypes (blue/yellow) and the colours of the
2073 colour map. We also made x and y scales uniform within each Figure. Finally, we removed the arrows from the
2074 the comments in Figure 5, to make the plots clearer.

REVIEWER 3

2075

2076 This manuscript reports results on (versions of) a mathematical / computational model. It aims to study the
 2077 effects of several aspects of motility on the rate of antibiotic resistance evolution in an environment that contains a
 2078 concentration gradient of an antibiotic.

2079 The model used is based on the so-called Staircase Model published in 2012 in PNAS (Hermsen *et al*), which
 2080 assumes a one-dimensional environment that is subdivided into compartments with increasing drug concentrations.
 2081 In the article, the analysis of the model was restricted to a particular parameter regime: it assumed that the carrying
 2082 capacity was large, the migration rate between neighboring compartments low, and mutation rates lower still. The
 2083 current manuscript expands this analysis by studying the regime of fast migration/motility. Next, the authors expand
 2084 the model by including stochastic switching between a fast and a slow motility state, or a density-dependent motility
 2085 rate.

2086 As I see it, the main findings are presented in the first four figures. Figure 1 demonstrates that that the behavior of
 2087 the high-motility regime is qualitatively different from that of the low-motility regime: it results in smoother density
 2088 profiles, which reduces the growth rate of mutants with a higher level of resistant emerging in the system and hence
 2089 increases the “ecological waiting time”. Figure 2 shows that, as a consequence, the adaptation rate depends non-
 2090 monotonously on the motility rate, with a positive dependence in the low motility regime but a negative dependence
 2091 in the high-motility regime. Figure 3 presents results of the model in which individual cells switch stochastically
 2092 between two motility rates. The main point is that this extended model behaves very similarly to the original model
 2093 with some effective migration rate. The same holds for the model with density-dependent motility, as presented in
 2094 Figure 4.

2095 Apart from these main points presented in full in the main text, quite a lot of additional work has been shifted to
 2096 supplementary figures, where the effect of a fitness cost, chemotaxis, HGT, different types of antibiotics, etc. are also
 2097 examined, and to the supplementary text, which contains detailed mathematical derivations and analysis.

2098 Generally, I think the analysis is sound and the conclusions seem warranted. The text is clearly and attractively
 2099 written and the figures are carefully designed. I only have minor comments.

2100

A4. Comments on content

2101 1. Around the same time when the Staircase Model was published, Greulich *et al* (PRL, 2012) published a similar
 2102 model. An important difference between Hermsen *et al* and Greulich *et al* was that Hermsen *et al* focused on a series
 2103 of relatively isolated compartments, whereas Greulich *et al* considered continuous space. It seems to me that this
 2104 difference disappears in the high-motility regime. While the authors cite Greulich *et al*, they do not compare their
 2105 results to that paper in much detail. I think they should.

2106 Greulich and colleagues focus on the shape of the antibiotic gradient and various mutational pathways rather than
 2107 the effect of cell motility on the evolution of antibiotic resistance. However, some of their results can be directly
 2108 compared with our work. For example, the authors report that “for the nonuniform drug distribution [Fig. 2(b)], t
 2109 (the average time to resistance) varies non-monotonically as a function of the (gradient) steepness, with a minimum
 2110 at 0.01” and this result is comparable to the results in our Fig. 2b if the gradient steepness is replaced by motility.
 2111 To clarify, if motility increases, bacteria explore the steepness of the gradient faster, which explains the link between
 2112 our results and theirs. As we discuss the relationship between motility and gradient shape already by comparing our
 2113 work with [A15, A16], we now added the work of Greulich *et al* to this list of studies.

2114

2115

2116 2. Section D and Figure 5 of the Results section discusses the effect of migration on the density profile from a
 2117 dynamic systems perspective. Although the mathematics presented here and in the SI are fun (for some readers), the
 2118 ultimate biological messages are very intuitive and can easily be summarized in a few sentences, as the authors in fact
 2119 do in the last paragraph of this section. I would advice to move this part to the supplement.

2120 We agree that the biological messages of Section D and Figure 5 of the Results section are relatively intuitive,
 2121 but this section has an important purpose, to bring together key ideas from all variants of the staircase model. The
 2122 concept of density profiles is a crucial idea that runs through the entire paper. In section A, the density profiles
 2123 distinguish the low and high motility regimes and give rise to mutant fitness profiles. In section B, it is discussed how
 2124 the combination of density profiles and mutant fitness profiles can be used to understand the evolutionary dynamics,
 2125 namely the adaptation rate. In section C, the density profiles are used to understand the impact of phenotypic
 2126 switching on the evolutionary dynamics. Therefore, section D primarily serves as a section that connects ideas from
 2127 previous sections. The loss of density profiles at a critical motility is therefore a particularly important feature that
 2128 arises in all variants of the model (Fig. 2, 3, 4). We have now also added a new movie where we compare the dynamics

2129 of the basic staircase model (without phenotypic heterogeneity), the model with stochastic switching and the model
 2130 with density-dependent switching in the deadly motility regime (Supplementary Movie 5), which highlights their
 2131 similar behaviour for the same effective motility. Moreover, not only is it relevant to understand why the concept of
 2132 critical motility is robust across our models, but also it connects our work with other published models (e.g., ecology
 2133 literature). For example, the loss of genetic diversity in ecological models is also often associated with critical motility
 2134 [A17–A19], and the framework presented in section D provides an intuitive explanation of how this happens. For
 2135 these reasons, we are very keen to keep Section D in the main text.

2136

2137

2138 3. How did the authors calculate the adaptation rates of, say, Fig. 2b in their simulations? If I understand well,
 2139 they used the equation $a_R = 1/(\mathbb{E}T_{evo} + \mathbb{E}T_{eco})$ and measured T_{evo} and T_{eco} in their simulations. I would argue
 2140 it makes more sense to measure the actual time it took in the simulations to take a certain number of steps. The
 2141 authors argue that this should equal the equation above, but this is only approximately so. In the current setup, this
 2142 assumption remains untested.

2143 Thank you for this question because it made us realize that we should clarify an important aspect of our methodology.
 2144 We measured the actual time it took in the simulations to take a certain number of steps, and we followed closely
 2145 the definition of a step used in previous work [A11]: “An adaptation step will occur when a founder, with genotype
 2146 7, finds compartment 7.” In particular, in each simulation, we noted when a first mutant $g = R + 1$ appeared in the
 2147 overlap region $x = R + 1$ for each resistance state R . Let us call these the founder states at resistance R (Fig. A5a).
 2148 The time between two founder states R and $R + 1$ can be split by an intermediate time when mutants $g = R + 1$
 2149 outcompete wild-type $g = R$ in the overlap region $x = R + 1$. Let us call these the winner states at resistance $R + 1$
 2150 (Fig. A5a). To avoid confusion, we now added upper indices R on the waiting times and mentioned founder and
 2151 winner states in Results section A. With this new notation, the time before the winner state is the ecological time
 2152 T_{eco}^{R-1} and the time after the winner state is the evolutionary time T_{evo}^R (Fig. A5a). Therefore, the time measured in
 2153 the simulations is precisely the sum of the ecological and evolutionary time, and the definition of the adaptation rate
 2154 becomes $a_R = 1/(\mathbb{E}T_{eco}^{R-1} + \mathbb{E}T_{evo}^R)$. We now clarify the definition and estimation of the adaptation rate in Results
 2155 section A, Methods, a new panel (a) of Fig. 2, and a new Supplementary Fig. 1 (Fig. A5 here).

2156 While our initial methodology builds on previous work, the reviewer’s comment made us realise that the adaptation
 2157 rate can alternatively be calculated by measuring the time between consecutive winner states (Fig. A5b), rather
 2158 than consecutive founder states (Fig. A5a). In such case, $a_R = 1/(\mathbb{E}T_{evo}^R + \mathbb{E}T_{eco}^R)$. In [A11], the two definitions of
 2159 adaptation rate match as this work considered low motility only and $\mathbb{E}T_{eco}^R \approx 0$ at low motility; we now highlight this
 2160 idea in Results section A. However, at high motility, a precise definition of a step matters as $\mathbb{E}T_{eco}^{R-1} \neq \mathbb{E}T_{eco}^R \neq 0$.
 2161 Importantly, even if adaptation rate is calculated from winner states (Fig. A5b,c), we observe the same dynamics
 2162 we obtained when adaptation rate was calculated from founder states (Fig. 2b). Moreover, if the waiting times are
 2163 calculated for appropriate R , there is a match between simulations and analytics for the adaptation rate in Fig. A5c,
 2164 which provides a further test that adaptation rate a_R can be inferred from the waiting times T_{evo}^R and T_{eco}^R .

2165

2166

2167 4. In introduction and discussion you claim that “cell motility has largely been overlooked”. I think this is claim is
 2168 a bit dubious given that all mathematical models that you cite do include motility, albeit at a fixed and relatively low
 2169 rate. Perhaps you can make this claim more specific. The mathematical models cited consider a form of “motility”
 2170 that essentially represents passive transport between antibiotic concentrations, and not active motility as we consider
 2171 here. For example, [A11] considers the transmission of bacteria from one organ to another or from one patient to
 2172 another. In our work, we consider active motility, which allows us to study the effect of different types of motility
 2173 phenotypes that bacteria are known to display, namely biased motility, stochastic switching and density-dependent
 2174 switching. To avoid confusion, we now changed “cell motility has largely been overlooked” to “active motility of
 2175 bacteria remains poorly explored”

2176

A5. Comments on figures

2177 1. In several figures, genotype is indicated with a blue or yellow color scale. Even though genotype is restricted to
 2178 integers, the legend belonging to the color scale shows integers at the *border* between the colors. This is confusing.
 2179 We now changed the colours in heatmaps to avoid this confusion.

2180

2181

2182 2. In the three panels of Fig. 3b, the lines indicating the borders between the regimes seem to be inconsistent. In
 2183 particular, in the heat maps the horizontal and vertical black lines are not at $\nu_i = \delta = 10^{-1}$.

Figure A5 | Alternative definition of adaptation rate. (a) Adaptation rate between consecutive founder states. The adaptation process in the staircase model corresponds to jumps between resistance states $R \rightarrow R + 1$ that happen at rate a_R . The resistance states R are characterised by founder states (a first mutant appears in the overlap region) and winner states (mutants outgrow wild-type in the overlap region), which are separated in time by evolutionary T_{evo}^R and ecological T_{eco}^R waiting times. In the main text, we define the adaptation rate a_R as the rate at which consecutive founder states appear, which is a definition used in previous work [A11]. (b) Adaptation rate between consecutive winner states. As an alternative to adaptation rate between consecutive founder states, one can consider adaptation rate as the rate at which consecutive winner states appear. (c) Adaptation rate as a function of motility rate. The same description as in Fig. 2b, but the definition of adaptation rate used is according to winner states. Both definitions of adaptation rate lead to the same key results: adaptation rate increases at low motility, decreases at high motility, and no adaptation occurs at deadly motility. Moreover, simulations (dots) of the adaptation rate match appropriate analytical computation (lines).

2184 We now changed the lines in Fig. 3b to be consistent. However, notice that one of these lines describes the surface
 2185 of $\nu_1 + \nu_2 = 2\delta$, as indicated in the figure. In log-log plot, this surface has the shape of a line with a smooth corner
 2186 rather than a smooth line perpendicular to $\nu_1 = \nu_2$ (as is usual in plots with linear scale), which explains why these
 2187 lines end at $\nu_i = 2\delta$ rather than $\nu_i = \delta$ as suggested by the reviewer.

2188

2189

2190 3. I would prefer that the parameters used in the figures are mentioned in the figure caption.

2191 We understand the concern of the reviewer, and indeed we considered the suggested option initially. However, we
 2192 realized that this option would significantly increase the length of the caption of each figure. Our figures typically

2193 have several panels and parameters would need to be specified for each panel separately. As it can be appreciated
 2194 in our section dedicated to the description of parameters, Supplementary Text S10, such descriptions are lengthy.
 2195 Therefore, we favour the format originally submitted.

2196 A6. Comments on text and notation

2197 1. What is the rationale behind the term “overlap region”? Overlap between what?

2198 We use the term overlap region to denote the only spatial compartment where the regions where mutant cells can
 2199 grow and regions where wild-type cells cannot grow overlap. We modified the introduction of the term in Results
 2200 section A to make this idea clearer.

2201 We note that previous work used the term sink instead of overlap region [A11, A20]. The term sink comes from the
 2202 ecological literature where source/sink denotes a compartment with a positive/negative net growth rate, respectively.
 2203 In particular, the overlap region is the only sink occupied by the wild-type population when motility is low (Fig.
 2204 1d, Theorem 3). However, when motility is high, the wild-type occupies all sink compartments $x > R$ and the
 2205 terminology used in previous work that focused on low motility only [A11, A20] cannot be used for the overlap region
 2206 $x = R + 1$. Using the source/sink notation, the overlap region is the only compartment where the wild-type sink and
 2207 mutant source overlap.

2208
 2209 2. In several places (both in the main text and in the supplement), assumptions are made that are not explicitly
 2210 mentioned. As mentioned above, the equation $a_R = 1/(\mathbb{E}T_{evo} + \mathbb{E}T_{eco})$ is only approximately true. Also, I don't think
 2211 it is mentioned anywhere that the analytical results rely on the assumption that K is large (at least, I think they
 2212 do) and that the mutation rates are low. (Without these assumptions, I do not expect the mean-field equations and
 2213 quasi-steady-state assumptions to work.) Please check the text for implicit assumption.

2214 We now explicitly mentioned all assumptions in the main text. We note that as a reply to one of the questions of
 2215 reviewer 1, we now considered the effect of different mutation rates, and showed that higher mutations rates do not
 2216 affect the key conclusions of our work (Fig A2 here, new Supplementary Fig. 3). However, this comment made us
 2217 realize that we had implicitly assumed that the first mutant in the overlap region always survives. Therefore, we now
 2218 consider the mutant survival probability in our analytics (new Supplementary Text S8) and explicitly address this
 2219 assumption. While the incorporation of mutant survival probability did not affect our conclusions, it improved our
 2220 analytical results when the environment is sink-like and motility is close to the critical motility (Fig. 2b). We thank
 2221 the reviewer for this helpful comment that allowed us to make this improvement.

2222
 2223
 2224 3. The notation of the expected value in the equation $a_R = 1/(\mathbb{E}T_{evo} + \mathbb{E}T_{eco})$ in the main text is not introduced.

2225 We now explicitly mentioned the notation for expected values. Furthermore, we added upper indices on T_{evo} and
 2226 T_{eco} to clarify the resistance state R , as explained in Comments on content, question 3.

2227
 2228
 2229 4. On line 195, the authors write “low motility ($\nu < \delta$) accelerates the adaptation rate”. Similar phrases appear
 2230 later in the manuscript, including on line 216 (“high motility (...) decelerates the adaptation rate”). These phrases
 2231 are confusing because in fact the adaptation rate is much lower in the low-motility regime than in the high-motility
 2232 regime. I would rephrase this.

2233 The idea that heterogeneous environments can accelerate the evolution of antibiotic resistance has been introduced
 2234 and discussed in the literature, for example in [A5, A16, A21, A22], and we use the same terminology when discussing
 2235 the net effect of bacterial motility on bacterial adaptive evolution. We show that bacterial motility can increase the
 2236 adaptation rate in heterogenous environments at lower rates (i.e., accelerates bacterial evolution), consistent with the
 2237 ideas published in the literature. However, we wanted to highlight that for high rates of motility, motility can actually
 2238 decrease bacterial adaptation rate (i.e., decelerates bacterial evolution). It should be noted that we compare local
 2239 trends, within identical heterogeneous environments that differ infinitesimally in cell motility. We now highlight that
 2240 our comparison is local to avoid confusion whenever it is not clear if we refer to global or local trends.

2241
 2242
 2243 5. line 290: “to to”

2244 Changed.

2245
 2246
 2247 6. line 328: What is meant by “stochastic switching levels out the numbers of both phenotypes”?

2248 We meant that in each compartment, the ratio of phenotypes is driven to 1:1 by stochastic switching. We have now
2249 changed to: “large stochastic switching quickly equilibrates the abundance of both phenotypes”.

2250

2251

2252 7. line 479: “Similar optimum” – > “A similar optimum”

2253 Changed.

REFERENCES

- 2254
- 2255 [A1] Hastings, A. Can spatial variation alone lead to selection for dispersal? *Theoretical Population Biology* **24**, 244–251
2256 (1983).
- 2257 [A2] Köhler, T., Michea-Hamzhepour, M., Plesiat, P., Kahr, A.-L. & Pechere, J.-C. Differential selection of multidrug efflux
2258 systems by quinolones in pseudomonas aeruginosa. *Antimicrobial agents and chemotherapy* **41**, 2540–2543 (1997).
- 2259 [A3] Sharma, S. K. & Mohan, A. Multidrug-resistant tuberculosis: a menace that threatens to destabilize tuberculosis control.
2260 *Chest* **130**, 261–272 (2006).
- 2261 [A4] Kohanski, M. A., DePristo, M. A. & Collins, J. J. Sublethal antibiotic treatment leads to multidrug resistance via
2262 radical-induced mutagenesis. *Molecular cell* **37**, 311–320 (2010).
- 2263 [A5] Zhang, Q. *et al.* Acceleration of emergence of bacterial antibiotic resistance in connected microenvironments. *Science*
2264 **333**, 1764–1767 (2011).
- 2265 [A6] Baym, M. *et al.* Spatiotemporal microbial evolution on antibiotic landscapes. *Science* **353**, 1147–1151 (2016).
- 2266 [A7] Liu, Z. & Papadopoulos, K. D. Unidirectional motility of escherichia coli in restrictive capillaries. *Applied and environ-*
2267 *mental microbiology* **61**, 3567–3572 (1995).
- 2268 [A8] Kinoshita, Y. *et al.* Distinct chemotactic behavior in the original escherichia coli k-12 depending on forward-and-backward
2269 swimming, not on run-tumble movements. *Scientific Reports* **10**, 15887 (2020).
- 2270 [A9] Sloan, J. B. & Urban, J. E. Growth response of escherichia coli to nutritional shift-up: immediate division stimulation
2271 in slow-growing cells. *Journal of Bacteriology* **128**, 302–308 (1976).
- 2272 [A10] Michelsen, O., Teixeira de Mattos, M. J., Jensen, P. R. & Hansen, F. G. Precise determinations of c and d periods by
2273 flow cytometry in escherichia coli k-12 and b/r. *Microbiology* **149**, 1001–1010 (2003).
- 2274 [A11] Hermsen, R., Deris, J. & Hwa, T. On the rapidity of antibiotic resistance evolution facilitated by a concentration gradient.
2275 *Proceedings of the National Academy of Sciences* **109**, 10775 – 10780 (2012).
- 2276 [A12] Pope, C. F., McHugh, T. D. & Gillespie, S. H. Methods to determine fitness in bacteria. *Antibiotic Resistance Protocols:*
2277 *Second Edition* 113–121 (2010).
- 2278 [A13] Rousset, F. *Genetic structure and selection in subdivided populations*, vol. 40 (Princeton University Press, 2004).
- 2279 [A14] Bentley, M. *The dynamical systems theory of natural selection*. Ph.D. thesis, University of Oxford (2016).
- 2280 [A15] Hermsen, R. The adaptation rate of a quantitative trait in an environmental gradient. *Physical Biology* **13**, 065003
2281 (2016).
- 2282 [A16] Steel, H. & Papachristodoulou, A. The effect of spatiotemporal antibiotic inhomogeneities on the evolution of resistance.
2283 *Journal of theoretical biology* **486**, 110077 (2019).
- 2284 [A17] Bulmer, M. Multiple niche polymorphism. *The American Naturalist* **106**, 254–257 (1972).
- 2285 [A18] Holt, R. D. & Gomulkiewicz, R. How does immigration influence local adaptation? a reexamination of a familiar
2286 paradigm. *The American Naturalist* **149**, 563–572 (1997).
- 2287 [A19] Lenormand, T. Gene flow and the limits to natural selection. *Trends in Ecology & Evolution* **17**, 183–189 (2002).
- 2288 [A20] Hermsen, R. & Hwa, T. Sources and sinks: a stochastic model of evolution in heterogeneous environments. *Physical*
2289 *review letters* **105**, 248104 (2010).
- 2290 [A21] Greulich, P., Waclaw, B. & Allen, R. J. Mutational pathway determines whether drug gradients accelerate evolution of
2291 drug-resistant cells. *Physical review letters* **109**, 088101 (2012).
- 2292 [A22] De Jong, M. G. & Wood, K. B. Tuning spatial profiles of selection pressure to modulate the evolution of drug resistance.
2293 *Phys. Rev. Lett.* **120**, 238102 (2018).

Reviewers' Comments:

Reviewer #1:

Remarks to the Author:

I have read the response of the authors to the reviewers and I think the comments have been sufficiently addressed.

I therefore recommend acceptance of the manuscript.

Reviewer #2:

Remarks to the Author:

The authors have thoroughly answered all points I have raised, and solidified their claim with more complex models. Yet, I leave their answer to point 5 regarding the title to the attention of the editor, as I remain convinced that it would be informative for the reader to be well aware that the conclusions of this work are solely based on modelling.

Reviewer #3:

Remarks to the Author:

The authors have responded to all of my questions, and in most cases to my satisfaction.

In my earlier review, I indicated that my remarks were minor, and I meant it, so bickering about them feels a little belligerent; but I do think I should mention the following:

* One of my comments was on the color scale in the *legends* of Figures 1d, 3c, and 4c. This legend should indicate the color representing each genotype. Each genotype is assigned an integer, and hence the legend should show which color belongs to each integer. But the legend instead shows integers at the *border between* colors. In figure 3c, for instance, there are 4 colors but 5 integers. Which integer belongs to which color?

It seems that the authors have adjusted the color scale of other figures instead of fixing this issue with the legend.

* Despite my suggestion, the authors have decided not to include information about parameter values in the captions. I suppose it's a matter of opinion, but since parameter values differ between figures I find it hard to compare results in different figures without referring the supplementary text. For example, without the supplement, it is impossible to know that the value used for δ in Fig. 4c deviates from the value used in Fig. 3C.

In retrospect, I realize that confusion about the parameter values was also responsible for the inconsistency of the lines in previous version of Fig. 3b: probably, the left-most panel in that figure was based on $\delta = 0.1$, the others on $\delta = 0.3$. Nevertheless, the other figures indicated δ at the axes at value 0.1. Happily, the authors did correct this now.

Comments from reviewers

REVIEWER #1 (REMARKS TO THE AUTHOR):

I have read the response of the authors to the reviewers and I think the comments have been sufficiently addressed. I therefore recommend acceptance of the manuscript.

Thank you.

REVIEWER #2 (REMARKS TO THE AUTHOR):

The authors have thoroughly answered all points I have raised, and solidified their claim with more complex models. Yet, I leave their answer to point 5 regarding the title to the attention of the editor, as I remain convinced that it would be informative for the reader to be well aware that the conclusions of this work are solely based on modelling.

The title was now changed to highlight that bacterial motility can affect, and indeed govern, the dynamics of antibiotic resistance evolution, which is what our work effectively shows.

REVIEWER #3 (REMARKS TO THE AUTHOR):

The authors have responded to all of my questions, and in most cases to my satisfaction.

In my earlier review, I indicated that my remarks were minor, and I meant it, so bickering about them feels a little belligerent; but I do think I should mention the following:

* One of my comments was on the color scale in the *legends* of Figures 1d, 3c, and 4c. This legend should indicate the color representing each genotype. Each genotype is assigned an integer, and hence the legend should show which color belongs to each integer. But the legend instead shows integers at the *border between* colors. In figure 3c, for instance, there are 4 colors but 5 integers. Which integer belongs to which color?

It seems that the authors have adjusted the color scale of other figures instead of fixing this issue with the legend.

We have now changed the legends accordingly. We initially misunderstood the reviewer's concern as a concern regarding the similarity of the colours in panels 3b and 4b, rather than the position of legends in panels 3c and 4c.

* Despite my suggestion, the authors have decided not to include information about parameter values in the captions. I suppose it's a matter of opinion, but since parameter values differ between figures I find it hard to compare results in different figures without referring the supplementary text. For example, without the supplement, it is impossible to know that the value used for δ in Fig. 4c deviates from the value used in Fig. 3C.

In retrospect, I realize that confusion about the parameter values was also responsible for the inconsistency of the lines in previous version of Fig. 3b: probably, the left-most panel in that figure was based on $\delta = 0.1$, the others on $\delta = 0.3$. Nevertheless, the other figures indicated δ at the axes at value 0.1. Happily, the authors did correct this now.

In addition, we have now included a minimal set of parameters in the caption of each Figure, while we still include the full characterisation of parameters used in the study in Supplementary Text 10 for the interested reader.